# Dynamic metabolic exchange governs a marine algal-bacterial interaction

Einat Segev[1]*, Thomas P Wyche[2], Ki Hyun Kim[2†], Jörn Petersen[3], Claire Ellebrandt[3], Hera Vlamakis[1‡], Natasha Barteneva[4], Joseph N Paulson[5], Liraz Chai[1§], Jon Clardy[2], Roberto Kolter[1]*

[1]Department of Microbiology and Immunobiology, Harvard Medical School, Boston, United States; [2]Department of Biological Chemistry and Molecular Pharmacology, Harvard Medical School, Boston, United States; [3]Leibniz-Institut DSMZ-Deutsche Sammlung von Mikroorganismen und Zellkulturen GmbH, Braunschweig, Germany; [4]Program in Cellular and Molecular Medicine, Boston Children's Hospital, Harvard Medical School, Boston, United States; [5]Department of Biostatistics and Computational Biology, Dana-Farber Cancer Institute, Boston, United States

*For correspondence:
Einat_Segev@hms.harvard.edu
(ES); roberto_kolter@hms.harvard.edu (RK)

Present address: [†]School of Pharmacy, Sungkyunkwan University, Suwon, Republic of Korea; [‡]Broad Institute, Cambridge, United States; [§]Institute of Chemistry, The Hebrew University of Jerusalem, Jerusalem, Israel

**Abstract** *Emiliania huxleyi* is a model coccolithophore micro-alga that generates vast blooms in the ocean. Bacteria are not considered among the major factors influencing coccolithophore physiology. Here we show through a laboratory model system that the bacterium *Phaeobacter inhibens*, a well-studied member of the Roseobacter group, intimately interacts with *E. huxleyi*. While attached to the algal cell, bacteria initially promote algal growth but ultimately kill their algal host. Both algal growth enhancement and algal death are driven by the bacterially-produced phytohormone indole-3-acetic acid. Bacterial production of indole-3-acetic acid and attachment to algae are significantly increased by tryptophan, which is exuded from the algal cell. Algal death triggered by bacteria involves activation of pathways unique to oxidative stress response and programmed cell death. Our observations suggest that bacteria greatly influence the physiology and metabolism of *E. huxleyi*. Coccolithophore-bacteria interactions should be further studied in the environment to determine whether they impact micro-algal population dynamics on a global scale.

## Introduction

There are many microbes that have influenced Earth's biogeochemistry. Prime among these are the coccolithophores, a diverse group of unicellular marine algae of the haptophyte division. Because of their high abundance, these micro-algae are fundamental in the global oxygen, carbon, and sulfur cycles (*Balch et al., 1991*; *Beaufort et al., 2011*; *Simó, 2001*; *Field et al., 1998*). As a consequence of their photosynthetic capacity these algae, together with other phytoplankton, are responsible for nearly half of our planet's primary production (*Field et al., 1998*). Coccolithophore cells are usually surrounded by elaborate platelets made of crystalline calcium carbonate (calcite) referred to as coccoliths. During coccolith production, carbon dioxide is released and can escape from the ocean to the atmosphere (*Marsh, 2003*). More importantly, coccoliths serve as a carbon sink as they accumulate on the bottom of the oceans (*Sabine et al., 2004*). Coccolith production by *E. huxleyi* accounts for roughly 1/3 of the total marine calcium carbonate production (*Iglesias-Rodríguez et al., 2002*). Hence, coccolithophores play a complex role in the global carbon cycle.

*Emiliania huxleyi* is the most widespread coccolithophore in modern oceans, forming dense annual blooms (*Paasche, 2001*). The blooms can cover thousands of square kilometers of ocean surfaces and are easily detected by satellites due to the highly reflective nature of the coccoliths

**eLife digest** Microscopic algae that live in the ocean release countless tons of oxygen into the atmosphere each year. Widespread algae – known as coccolithophores – surround their little plant-like body with a mineral shell made of a material similar to chalk. These microscopic algae form seasonal blooms. Over several weeks in early summer, the algae grow to enormous numbers and cover hundreds of thousands of square kilometers in the ocean. These blooms become so vast that satellites can detect them. However, suddenly the blooms collapse; the algae die and their chalky shells sink to the bottom of the ocean where they have been accumulating for millions of years.

More and more evidence suggests that these tiny algae interact with bacteria in various ways. However, so far, no one had documented a direct interaction between bacteria and a member of this key group of algae.

Now, in a controlled laboratory environment, Segev et al. show that marine bacteria from the Roseobacter group physically attach onto a tiny coccolithophore alga called *Emiliania huxleyi*. While the bacteria are attached to their algal host, they enjoy a supply of nutrients that trickles from the algal cell. Unexpectedly, Segev et al. also discovered that the algae grow better in the presence of the bacteria. It turns out that the bacteria use a molecule that they obtain from their algal hosts to produce a small hormone-like molecule that in turn enhances the growth of the algae. However, after three weeks of growing together, the bacteria produce so much of the growth-enhancing molecule – which is harmful in higher concentrations – that they actually kill their algal host.

These findings suggest that the bacteria first promote the alga's growth to boost their supply of nutrients. But as algae grow older, the bacteria harvest the algae to enjoy a last pulse of nutrients and allow their offspring to swim away and attach to younger algae.

The next challenge will be to link these laboratory observations to the actual microbial interactions in the ocean. It will also be important to explore whether other algae and bacteria interact in similar ways and if bacteria contribute to the sudden collapse of algal blooms by killing the algae.

(*Balch et al., 1991*; *Holligan et al., 1983*). The blooms also exhibit unique dynamics; they form seasonally over several weeks and then suddenly collapse (*Behrenfeld and Boss, 2014*; *Lehahn et al., 2014*; *Tyrrell and Merico, 2004*), a process that has been attributed to viral infection (*Bratbak et al., 1993*; *Lehahn et al., 2014*; *Vardi et al., 2012*). Recent evidence suggests that environmental stresses and viral infection can trigger oxidative stress and a process similar to programmed cell death (PCD) in *E. huxleyi* (*Bidle et al., 2007*; *Vardi et al., 2009*; *Bidle, 2016*). The induction of PCD, which is an autocatalytic process, has been shown to occur in various widespread species of phytoplankton including *E. huxleyi*, and functional links have been demonstrated between viral infection, PCD, and algal bloom collapse (*Bidle, 2015*, *2016*; *Bidle and Vardi, 2011*; *Fulton et al., 2014*; *Vardi et al., 2009*, *2012*; *Rohwer and Thurber, 2009*). Interestingly, although *E. huxleyi* blooms harbor a rich community of bacteria, at times dominated by the Roseobacter group (*González et al., 2000*; *Green et al., 2015*), bacteria are not generally considered to be a factor influencing coccolithophore physiology and bloom dynamics.

Various types of phytoplankton were shown to have both mutualistic and antagonistic interactions with bacteria (*Amin et al., 2015*; *Miller and Belas, 2004*; *Miller et al., 2004*; *Wang et al., 2014*; *Durham et al., 2015*). In addition, the possible role of algicidal bacteria in the ocean has been examined and discussed (*Mayali and Azam, 2004*; *Harvey et al., 2016*). It has been previously suggested by our laboratories that bacteria might interact with *E. huxleyi* (*Seyedsayamdost et al., 2011*). However, coccolithophore-bacteria interactions have not yet been unambiguously demonstrated. This gap is curious because *E. huxleyi*'s important role in the global sulfur cycle is in part a consequence of an algal-bacterial interaction. *E. huxleyi* produces the osmolyte and antioxidant dimethylsulfoniopropionate (DMSP) (*Sunda et al., 2002*). This molecule, when released into the water by leakage or cell lysis, can be used by some bacteria as a source of sulfur and carbon (*Curson et al., 2011*; *González et al., 1999*). During DMSP catabolism, bacteria such as Roseobacters produce the volatile by-product dimethyl sulfide (DMS). *E. huxleyi* is also a producer of DMS, which is a bioactive gas

with possible roles in climate regulation (*Charlson et al., 1987*; *Alcolombri et al., 2015*).When DMS enters the atmosphere it is oxidized and serves to form cloud condensation nuclei (*Curson et al., 2011*; *González et al., 1999*). While the DMSP flux from algae to bacteria, and the production of DMS gas by both algae and bacteria have been clearly demonstrated, the role of DMS in climate regulation has been questioned (*Quinn and Bates, 2011*).

Accumulating evidence suggests that there may be widespread interactions between *E. huxleyi* and Roseobacters. *Phaeobacter inhibens* (*Buddruhs et al., 2013*), a well-studied member of the Roseobacter group, was shown to produce molecules that specifically affect *E. huxleyi* (*Seyedsayamdost et al., 2011*). This bacterium, when grown in a pure culture in the presence of p-coumaric acid, a product released by aging algae, produced novel compounds able to lyse *E. huxleyi*. The compounds were named roseobacticides and their discovery pointed towards a possible interaction between *P. inhibens* and *E. huxleyi* (*Seyedsayamdost et al., 2011*). Furthermore, we recently showed that lipid metabolism in *E. huxleyi* is altered in the presence of *P. inhibens* (*Segev et al., 2016*). However, a direct physical interaction between these algae and bacteria had not been previously described and no other details of their interaction were known. Here we describe the establishment of a co-culture model system between *E. huxleyi* and *P. inhibens* that allows the examination of the spatiotemporal dynamics of their interaction. We provide evidence that *E. huxleyi* and *P. inhibens* associate intimately when co-cultured. We show that bacteria promote algal growth but eventually kill their aging algal hosts. The same bacterial compound, indole-3-acetic acid, mediates stimulation of algal growth as well as algal death. Finally, algal death in the co-culture seems to involve an apoptotic process. Similar *E. huxleyi* - bacteria interactions might occur in the ocean and could thus affect algal physiology, bloom dynamics and biogeochemical cycles.

## Results and discussion

*E. huxleyi* and *P. inhibens* are two well-studied marine microbes. To determine if they co-occur in algal blooms we analyzed the bacterial community associated with *E. huxleyi* blooms using a culture-independent metagenomic approach. Two independent blooms were sampled in the Gulf of Maine during the summer of 2015. The results shown in *Figure 1* indicate that *P. inhibens* was indeed found co-occurring with *E. huxleyi* in algal blooms (*Figure 1*). Thus, the suggested interaction between these microorganisms might be ecologically significant. To study the interactions of *E. huxleyi* and *P. inhibens*, it was necessary to establish conditions to co-culture these two species. We started by examining pure cultures of each microorganism. Coccolith-forming (i.e. calcifying) *E. huxleyi* (strain CCMP3266) were inoculated into L1-Si, a seawater based medium supplemented with additional sources of phosphorus (0.04 mM $PO_4$), nitrogen (0.9 mM $NO_3$) and sulfur (0.08 μM $SO_4$), along with vitamins and trace metals (*Guillard and Hargraves, 1993*) (see Materials and methods). In this medium, *E. huxleyi* grows to $3 \times 10^5$ cell/ml. Under these conditions *E. huxleyi* produces calcium carbonate coccoliths that surround the algal cell (*Figure 2a*). *P. inhibens* DSM17395 is normally grown in the rich medium 1/2YTSS (*Seyedsayamdost et al., 2011*) (see Materials and methods) where it easily aggregates; it often forms 'rosette' structures through a polysaccharide-containing pole (*Figure 2b,c*) (*Segev et al., 2015*). Of note, alone these bacteria do not grow in the L1-Si medium (*Figure 2d*, grey bars). However, we found that bacteria do grow in co-culture with *E. huxleyi*. To grow a co-culture, we inoculated algae into L1-Si medium and, after four days, introduced bacteria into the algal culture. In these co-cultures, bacterial numbers increased nearly five orders of magnitude over a period of 14 days (*Figure 2d*, green bars). Microscopic examination of the co-culture revealed that some algae were no longer surrounded by coccoliths (*Figure 2e*). Rather, naked algal cells were now covered by bacteria attached via their poles. This attachment was evident in both fixed (*Figure 2e*) and live (*Figure 2f*) samples. Of note, attachment of *P. inhibens* to other micro-algae as well as macro-algae has been previously demonstrated (*Frank et al., 2015*). Using a specific fluorescent lectin to detect the polar polysaccharide (see Materials and methods), it appeared that bacterial attachment onto the algal cell also involves the polar bacterial polysaccharide (*Figure 2f*). Examination of co-cultures revealed that over time more algae have attached bacteria (*Figure 2g*) and each algal cell is associated with increasing numbers of bacteria as the co-culture ages (*Figure 2f,h*).

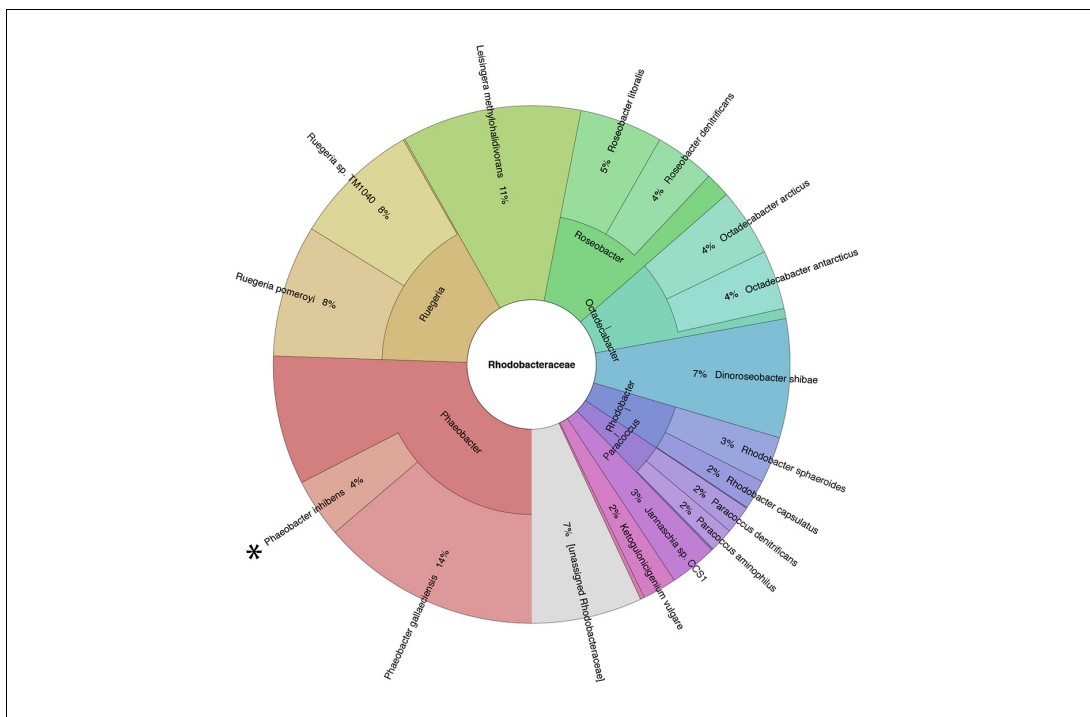

**Figure 1.** Metagenomic analysis of Roseobacters associated with *E. huxleyi* blooms reveals co-occurrence of *P. inhibens*. Two *E. huxleyi* blooms were sampled in the Gulf of Maine during the summer of 2015 and metagenomic analysis of the bacterial population was performed (see Materials and methods). Shown is the relative abundance of members of the Rhodobacteraceae family, which accounted for 6% of bacteria. The same members of the Rhodobacteraceae family were detected in both blooms and their abundance changed ±2% between replicates and between the two blooms. *P. inhibens* was present in both blooms and is indicated by an asterisk. Shown are the results for the July 2015 bloom (see Materials and methods).

Bacteria clearly benefit from interacting with the algal host as their growth is enabled by the algae in an otherwise non-permissive medium (*Figure 2d*). What do bacteria receive from algae that allows them to grow? Given that L1-Si does not contain significant amounts of organic carbon to permit robust growth of the heterotrophic bacteria, it stands to reason that the key nutrient that algae provide is fixed carbon. If indeed fixed carbon were to be the sole nutrient needed by the bacteria, addition of a utilizable form of carbon to L1-Si should enable bacterial growth. However, addition of 5.5 mM glucose did not lead to significant bacterial growth (*Figure 3a*). This was an unexpected result because, as mentioned above, L1-Si in addition to seawater also contains added phosphorus (0.04 mM $PO_4$), nitrogen (0.9 mM $NO_3$) and sulfur (0.08 µM $SO_4$). In fact, even addition of higher nutrient concentrations in forms shown to be utilizable by *P. inhibens* (*Zech et al., 2009*) as individual supplements (nitrogen 5 mM $NH_4$, phosphorous 2 mM $PO_4$, and sulfur 33 mM $SO_4$) or in various combinations of two or three of them did not lead to robust bacterial growth (*Figure 3a*). Only addition of all four essential nutrients resulted in bacterial growth to a density of $5 \times 10^8$ CFU/ml, which we normalized to 100% in *Figure 3a*. Thus, *E. huxleyi* can provide all four essential nutrients (C, N, P and S) in suitable forms and concentrations to enable growth of the heterotrophic bacterium *P. inhibens*.

The sulfur flux from algae to bacteria is of special interest. Because of its ecological importance, we wanted to investigate whether DMSP plays a role in the interaction between *E. huxleyi* and *P. inhibens*. First, we examined the ability of bacteria to grow with DMSP as a sole source of sulfur or carbon. As shown in *Figure 3a*, DMSP can serve as a sulfur source (*Figure 3a*, 'CNP + DMSP 30 µM'). In contrast, DMSP does not supply sufficient carbon to support robust bacterial growth. (*Figure 3a* 'NP + DMSP 30 µM'). Even when DMSP was added in higher concentrations to supply carbon in a comparable amount to the carbon supplied by the 5.5 mM glucose in the parallel

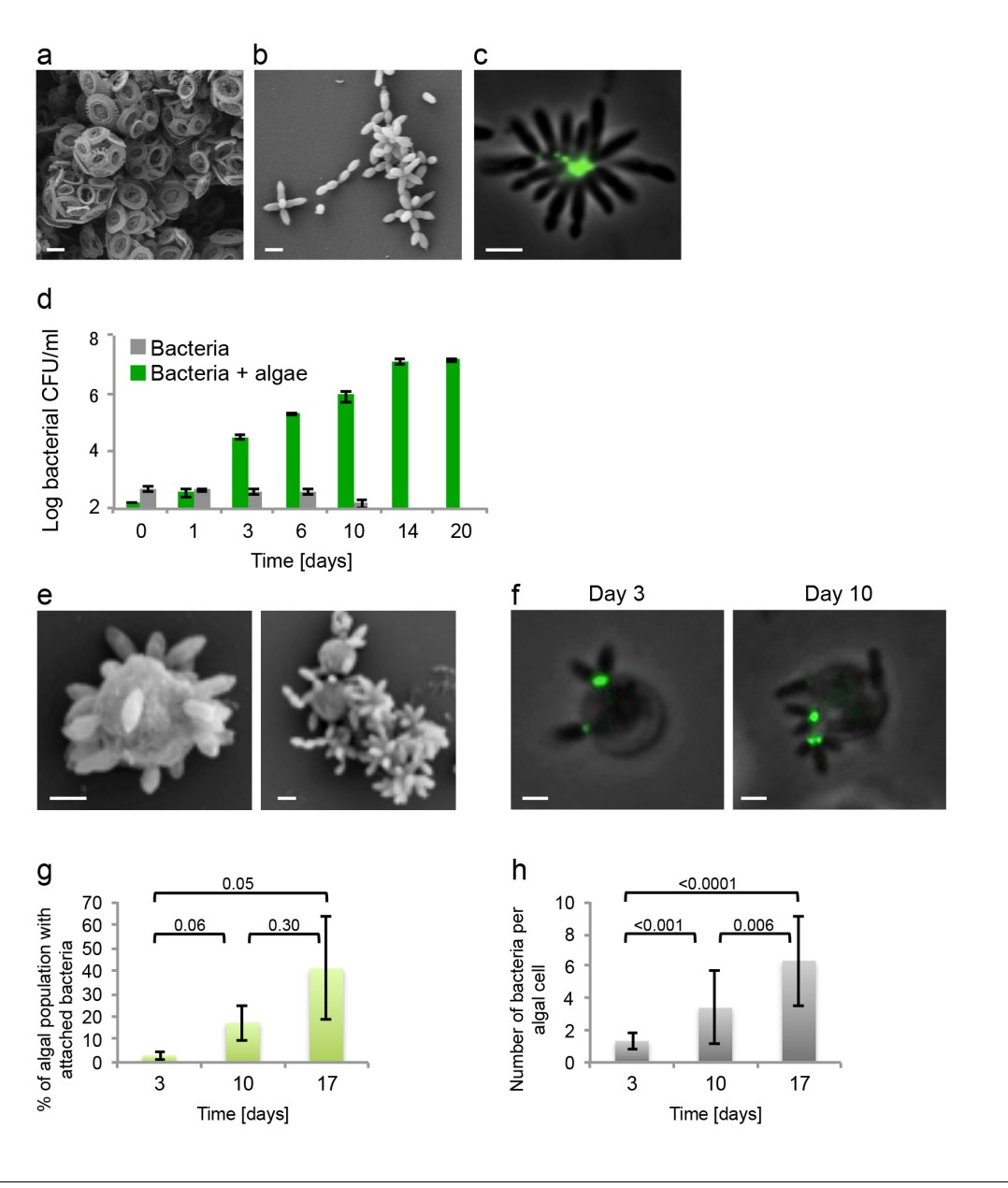

**Figure 2.** Algal-bacterial co-cultures. (a) Scanning electron microscopy (SEM) image of *E. huxleyi* (CCMP3266) pure algal culture. (b) SEM image of *P. inhibens* (DMS17395) pure bacterial culture. (c) Overlay image of a pure culture of *P. inhibens* bacteria (phase contrast microscopy, grey) stained with a fluorescent lectin (Alexa Fluor 488 conjugated lectin, green). (d) Bacteria grown in L1-Si medium in the absence (grey bars) and presence (green bars) of algae over 20 days. Error bars represent the standard deviation of two biological replicates. (e) SEM image of cells from an algal-bacterial co-culture. (f) Phase contrast microscopy imaging of live co-culture samples (grey) overlaid with images of the fluorescent lectin (Alexa Fluor 488 conjugated lectin, green) showing increasing numbers of bacteria attaching onto algal cells over time. (g) Quantification of algal cells with attached bacteria as a function of time, n > 300. Error bars represent the standard deviation between the multiple examined fields. (h) Quantification of the number of attached bacteria per algal cell as a function of time, n > 300. Error bars represent the standard deviation between the multiple examined fields. All scale bars in the figure correspond to 1 μm. Statistical significance was calculated using a Student's T-test and *p* values are presented above datasets.

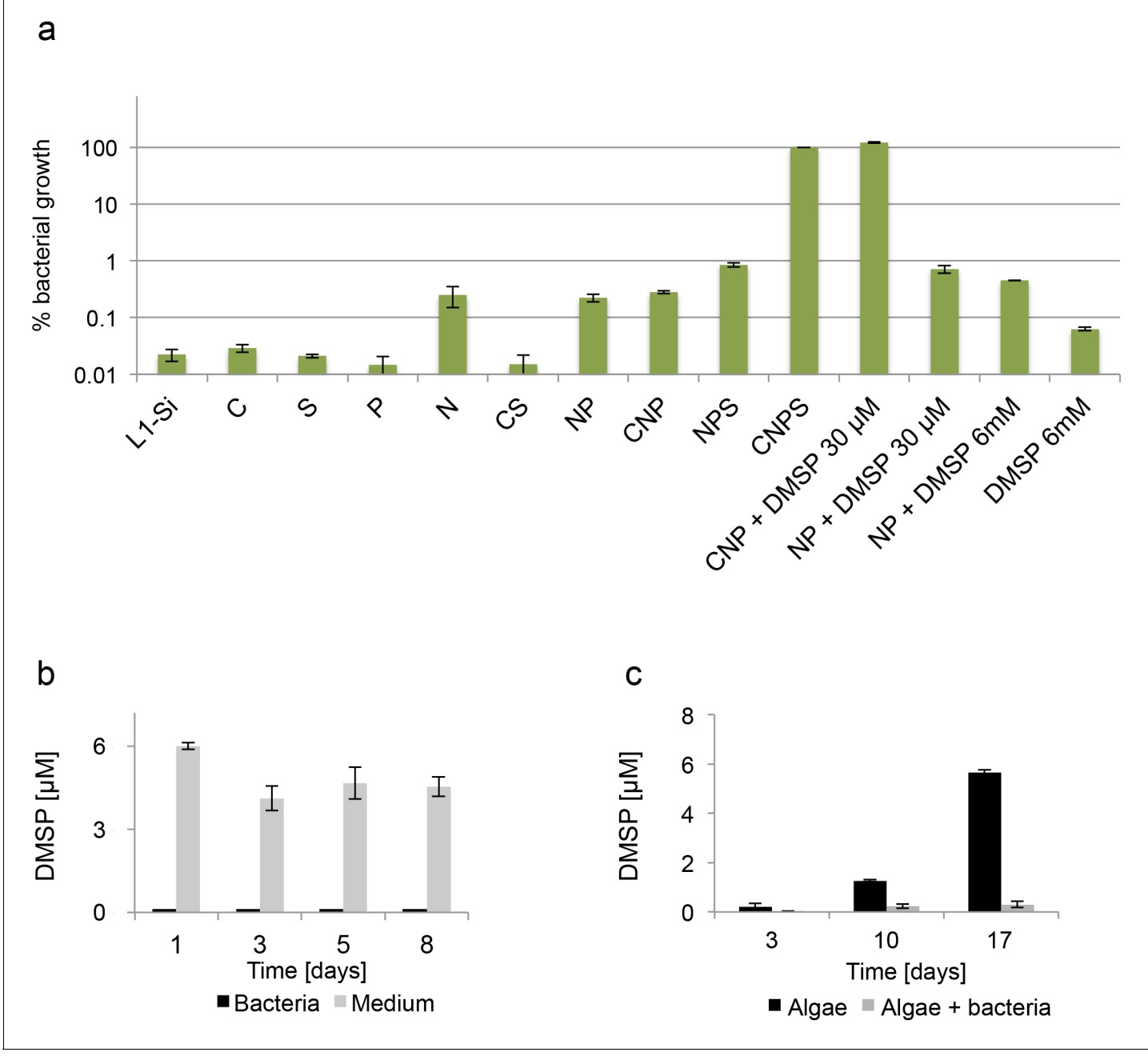

**Figure 3.** Bacteria require essential nutrients to grow in L1-Si. (a) Bacterial growth in L1-Si medium supplemented with various essential nutrients (C-glucose, N-nitrogen, P-phosphorus, S-sulfur) was monitored over eight days (see Materials and methods). Presented are the maximal growth values that were obtained after three days of incubation. The initial bacterial inoculum was $1 \times 10^5$ CFU/ml. Growth in CNPS reached $5 \times 10^8$ and was normalized to 100%. (b) Bacteria consume externally added DMSP. (c) DMSP production by *E. huxleyi* in pure culture (black bars) and in co-culture (grey bars). Error bars represent the standard deviation between two biological replicates.

experiments, bacterial growth was not evident (*Figure 3a* 'NP + DMSP 6mM'). Next, we directly monitored DMSP consumption in a growing bacterial culture. Our measurements indicate that all of the added DMSP is rapidly utilized by the growing bacteria whereas in un-inoculated medium the DMSP levels remain relatively stable over time (*Figure 3b*). Based on these observations, we then proceeded to determine whether DMSP is produced and exuded by algae in pure culture and in co-cultures. Indeed, over a period of 17 days, DMSP concentration in the medium of a pure algal

culture increased, reaching a concentration of nearly 6 µM (*Figure 3c*, black bars). In contrast, in a co-culture of algae and bacteria, DMSP levels in the medium were nearly undetectable, presumably due to its rapid metabolism by the bacteria (*Figure 3c*, grey bars). Since bacteria are directly attached to their algal host it is possible that they experience a considerably higher local concentration of DMSP than the concentration measured in the bulk medium. It has been reported that Roseobacters and other bacteria can chemotax towards DMSP and catabolize it and various metabolic pathways for the bacterial use of the DMSP sulfur have been proposed and tracked (*Miller and Belas, 2004*; *Miller et al., 2004*; *Seymour et al., 2010*; *Brock et al., 2013*; *Wang et al., 2016*). Thus, it is possible that DMSP serves as a chemical cue attracting bacteria to colonize the *E. huxleyi* host cell. Additional experiments are clearly needed to determine if indeed DMSP serves as an infochemical promoting bacterial colonization. Yet, it has been previously shown that algal DMSP and exudates serve as a strong cue to attract bacteria (*Seymour et al., 2010*; *Smriga et al., 2016*).

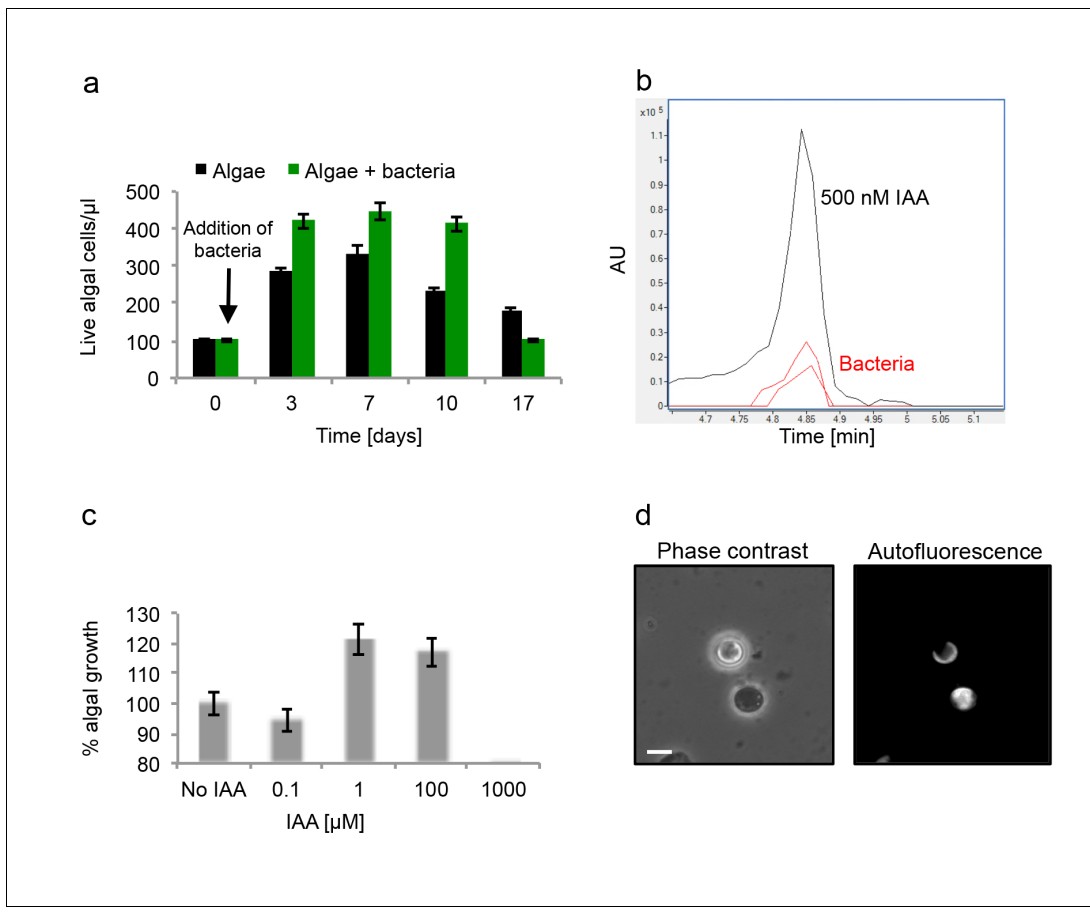

**Figure 4.** Algal growth in the presence of bacteria. (a) Algal growth was monitored over 17 days in the presence of bacteria (green bars) and in pure culture (black bars). (b) Indole-3-acetic acid (IAA) production was observed in pure bacterial cultures grown in L1-Si supplemented with essential nutrients. Shown is an LC-MS extracted ion chromatogram (EIC, *m/z* 176.0706 ± 10 ppm) of an IAA standard of 500 nM (black) and two biological replicates (red) (see Materials and methods). AU=Arbitrary Units. (c) Algal growth was examined upon addition of the auxin IAA. Percent algal growth is relative to a culture with no IAA added. Note that at a concentration of 1000 µM IAA, cell numbers dropped to less than 10%. (d) Following treatment with 1 mM IAA, the autofluorescence signal of dead algal cells (the lower cell in this image) appears similar to the signal observed in bacterially induced death (compare with *Figure 6I*), indicative of chloroplast deformation but partially intact cell membrane. Scale bar corresponds to 4 µm. Error bars in a and c represent the standard deviation between two biological replicates.

The following figure supplement is available for figure 4:

**Figure supplement 1.** Validating the detection of bacterially produced IAA.

To further characterize the algal-bacterial interaction we explored the bacterial effect on algal growth. Flow cytometry is commonly used to monitor algal growth. However, due to bacterial attachment and algal clumping, our co-cultures consist of aggregates of cells of varying sizes. Thus, it is challenging to accurately interpret the results of flow cytometry analyses. Therefore, our analyses included traditional flow cytometry as well as validation of our results using imaging cytometry in order to characterize the different particles (see Materials and methods). We found that during the initial 10 days of culturing, there were greater numbers of algae in the co-cultures compared to pure algal cultures (*Figure 4a*). In the co-culture, algae reach approximately 25% higher numbers in comparison with the maximum reached in the absence of bacteria. In addition, in an algal pure culture the death phase starts after day seven while in a co-culture a significant decrease in population is evident only at day 17. Interestingly, the death phase in an algal pure culture seems more gradual than the rapid demise observed in a co-culture (*Figure 4a*). Thus, it seems that the algal-bacterial interaction is dynamic. Initially the interaction is mutualistic, however over time bacteria may become harmful for their algal hosts.

Various bacteria that interact with plants are known to provide phytohormones (auxins) that promote plant growth (*Costacurta and Vanderleyden, 1995*). It was previously suggested that phenylacetic acid (PAA) produced by *P. inhibens* might serve as an auxin to enhance algal growth (*Seyedsayamdost et al., 2011*). We could not detect PAA in bacterial cultures or algal-bacterial co-cultures.

Intriguingly, the auxin indole-3-acetic acid (IAA) was previously demonstrated to play significant roles in numerous terrestrial plant-bacteria interactions and was recently shown to be key in a marine association between a diatom and a Roseobacter bacterium (*Amin et al., 2015*; *Spaepen et al., 2007*). As many members of the Roseobacter group have multiple metabolic pathways for IAA synthesis (*Moran et al., 2007*), we posited that IAA might be produced by *P. inhibens* to promote the growth of *E. huxleyi* in co-culture. To test this, we examined whether IAA is produced by *P. inhibens*. We were able to detect IAA (0.4 nM) in bacterial pure cultures (see Materials and methods) (*Figure 4b*). A standard of IAA was analyzed by high resolution LC-MS and targeted MS/MS (see Materials and methods). The retention time (*Figure 4b*), exact mass (*Figure 4—figure supplement 1a*), and fragmentation pattern (*Figure 4—figure supplement 1b*) of the standard matched with the IAA that was detected in bacterial cultures. Importantly, no IAA was detected in axenic algal cultures. Next we assessed algal growth upon addition of IAA. Similar to the increase we observed in co-culture, algal growth yield was improved by 20% upon addition of 1 μM and 100 μM IAA (*Figure 4c*).

Our attempts to monitor IAA levels in co-cultures revealed that in these conditions IAA was undetectable, suggesting rapid up-take by algae. This observation is in agreement with previous reports of significant binding of IAA and rapid signal transduction in plants, indicative of the high affinity towards this compound (*Kepinski and Leyser, 2005*). Interestingly, studies have described microbial signaling pathways taking place in the rhizosphere (the soil immediately surrounding the roots of terrestrial plants) (*Spaepen et al., 2007*). It is tempting to hypothesize that similar short-circuit processes could take place in the phycosphere (the immediate volume in proximity to the algal cell). Thus, it seems that in our experimental system bacteria inhabiting the algal phycosphere supply their algal host with growth promoting molecules.

To further characterize bacterial IAA production in the algal phycosphere, we examined whether similarly to other IAA-producing bacteria, *P. inhibens* will alter IAA production in response to exogenous tryptophan (*Brandl and Lindow, 1996*; *Patten and Glick, 2002*; *Prinsen, 1993*; *Theunis et al., 2004*; *Zimmer et al., 1998*). Indeed, addition of 0.1 mM tryptophan resulted in the production of 10.8 nM IAA, approximately 25-fold increase in comparison to conditions with no added tryptophan (*Figure 5a*). The increase in IAA production could be the result of two different mechanisms. Added tryptophan could enhance bacterial metabolism, thus resulting in general increase of bacterially-produced metabolites. Alternatively, exogenous tryptophan could be shunted primarily towards IAA production. To distinguish between these two possibilities, we supplemented a bacterial culture with uniformly labeled tryptophan ($^{13}$C and $^{15}$N). If the labeled tryptophan were taken up and directly shuttled towards IAA production, all IAA should be fully labeled (*m/z* 187) (*Figure 5b*). However, if the imported tryptophan participates in other cellular processes, and atoms are exchanged with endogenous pools of carbon and nitrogen, then the resulting IAA will not be fully labeled, thus resulting in a lower mass. The lowest possible mass would be of a fully

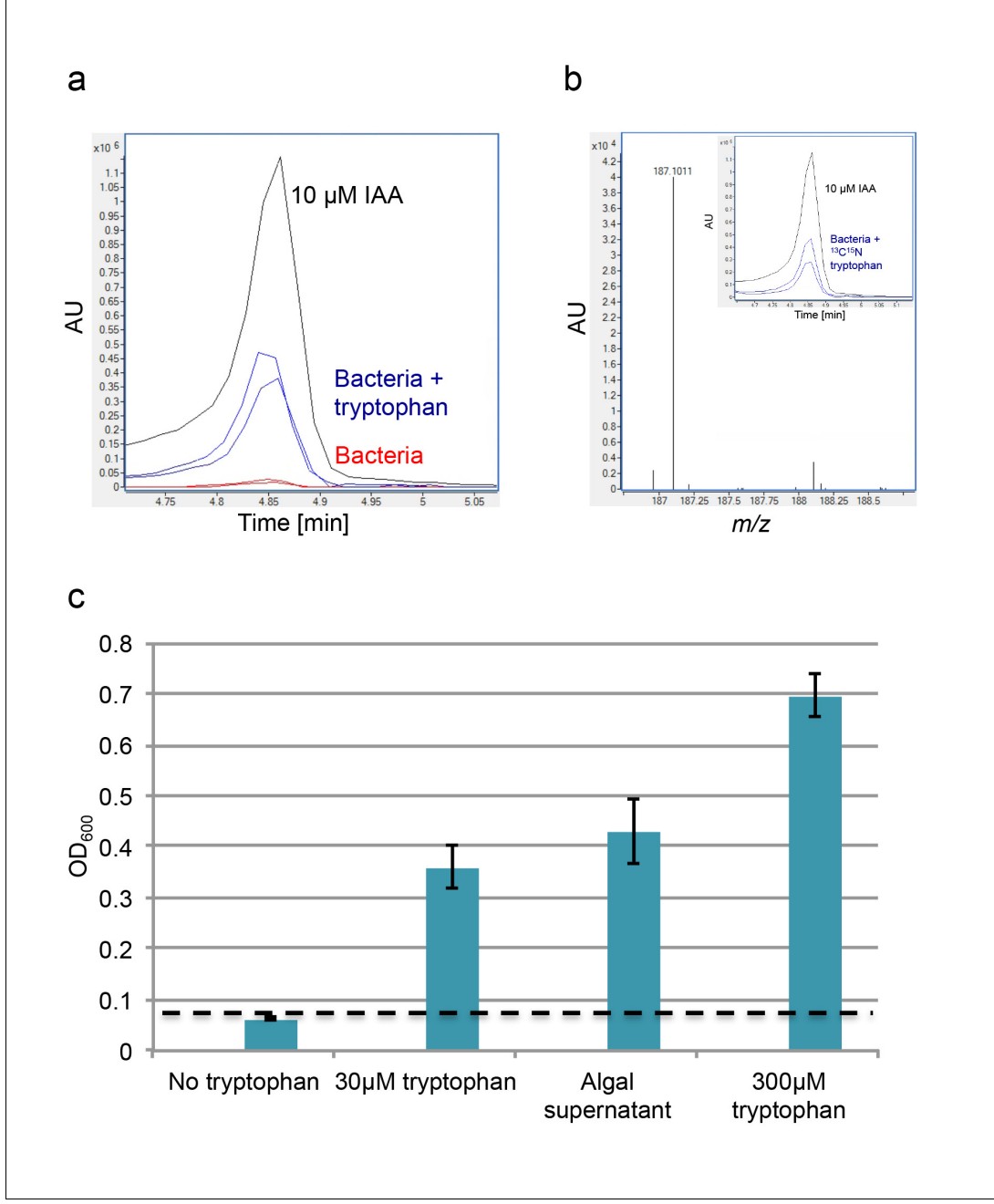

**Figure 5.** Exogenous tryptophan promotes bacterial IAA biosynthesis. (**a**) Addition of 0.1 mM tryptophan to *P. inhibens* cultures results in approximately 25-fold increase in produced IAA. Shown is an LC-MS extracted ion chromatogram (EIC, *m/z* 176.0706 ± 10 ppm) of an IAA standard of 10 μM (black), IAA detected in bacterial cultures supplemented with tryptophan (blue), and IAA detected in untreated bacterial cultures (red). (**b**) The addition of isotopically labeled tryptophan ($^{13}$C and $^{15}$N) leads to bacterial production of IAA with full isotopic incorporation, indicated by *m/z* 187.0011. Inset shows the LC-MS chromatogram of an IAA standard (black) (EIC, *m/z* 176.0706 ± 10 ppm) and the labeled IAA detected in two biological replicates (blue) (EIC, *m/z* 187.0011 ± 10 ppm). (**c**) Cell density ($OD_{600}$) after 16 hr at 30°C of a tryptophan auxotroph *E. coli* strain grown in known concentrations of tryptophan and in algal supernatant (see Materials and methods). Dashed line indicates density of the initial inoculum. Error bars represent the standard deviation between four biological replicates.

unlabeled IAA molecule (*m/z* 176). The results of our experiments indicate that all produced IAA is fully labeled (*Figure 5b*), thus indicating that exogenous tryptophan is directly converted to IAA. Of note, a recent study reported on the production of IAA in the mM range in axenic *E. huxleyi* cultures supplemented with tryptophan (*Labeeuw et al., 2016*). As all of our experiments and controls indicated no IAA production by algae, currently we do not understand the source of this apparent contradiction.

To elucidate the relevance of increased IAA production by bacteria in the presence of exogenous tryptophan, we wanted to examine whether *E. huxleyi* exudes tryptophan. If indeed tryptophan is exuded by algal cells, bacteria in the phycosphere could import it and convert it to IAA. Secreted bacterial IAA could be utilized by algae and lead to improved algal yields. Such chemical cross talk will feed into a positive algal-bacterial feedback loop. In line with this idea, in another algal-bacterial interaction it has been shown that tryptophan released by a diatom fuels IAA production by a bacterium (*Amin et al., 2015*). To monitor bioavailable tryptophan in *E. huxleyi* exudates, we used an *Escherichia coli* strain that is a strict tryptophan auxotroph. This bacterial strain relies solely on exogenous tryptophan; in the presence of extracellular tryptophan the strain will grow and in the absence of tryptophan it will not grow. Cultivation of this strain in minimal medium supplemented with filtered algal supernatant resulted in marked bacterial growth (*Figure 5c*). This indicates that algal exudates contain bioavailable tryptophan. Interestingly, we detected tryptophan (100 nM) in filtered samples of *E. huxleyi* blooms (see Materials and methods), suggesting that this metabolite may be relevant to environmental interactions.

To better understand the temporal dynamics of the algal-bacterial interaction we examined co-cultures over a period of 17 days and compared them with algal pure cultures of the same age. After 10 days of incubation, no visible difference was apparent between algal pure cultures and co-cultures (*Figure 6a,b*). However, after 17 days of incubation the color of the co-cultures rapidly changed from green (the color of healthy algae) to white, a process we refer to as 'bleaching' (*Figure 6c,d* and *Video 1*). To better understand these algal changes we examined the various cultures under the microscope. This revealed that the algal cells in a bleached co-culture exhibit more coccolith shedding (*Figure 6e,f,g,h*, note arrow in *Figure 6h*). Furthermore, the typical fluorescent crescent shape that is made of the two chloroplasts in each cell (*Figure 6i,j,k*) is lost in the bleached co-culture and the fluorescent signal emanating from chlorophyll and accessory pigments fills the entire cell (*Figure 6l*). In addition, using a viability stain (Sytox green, see Materials and methods) it is evident that the aging co-culture contains mostly dead algal cells (*Figure 6m,n,o,p*). One possible explanation for the bleaching observed is that the bacteria simply degrade dying algal cells at day 17 while similarly dying algal cells in the pure culture at the same time point remain intact. However, comparison of the death rates in the algal population in pure and co-cultures reveals a significant difference. At this time point, the vast majority of algal cells in the co-culture are dead (94% as indicated by Sytox staining, see *Figure 3p*) while in the pure culture only 21% are dead by day 17. Thus, the presence of bacteria seems to play a key role in promoting algal death. Taken together, these results suggest that following the mutualistic phase in the algal-bacterial interaction, bacteria become pathogens that cause the bleaching and death of their algal partners.

To further explore the death process experienced by algae in co-culture, we examined the expression profile of a select group of algal genes. Out of 50 examined genes representing various functional groups (*Supplementary file 1*), 15 genes exhibited significant up-regulation in algae over time in co-culture (*Table 1*). To test whether this group of 15 genes exhibits expression levels that are significantly higher than the other 35 genes (in *Supplementary file 1*), we first calculated the variance in the two datasets at day 17 using an F-test. The variance was not found to be statistically different and thus a two-tailed Student's T-test was performed assuming two samples with equal variance. The resulting probability of the T-test was $2.8 \times 10^{-11}$ indicating that the difference between the two datasets is statistically significant. The highly up-regulated genes encode proteins that are presumed to be involved in various aspects of oxidative stress and programmed cell death (PCD). PCD naturally occurs in an aging algal population and can be triggered by a variety of biotic stresses such as viral infection and abiotic environmental stresses such as nutrient limitation and various light regimes (*Bidle, 2015*; *2016*). Our expression data suggest that PCD as well as responses to oxidative stress are more prevalent among algal cells in co-culture. Thus, PCD in *E. huxleyi* seems to be triggered by bacteria. Of note, the up-regulation of genes encoding proteins

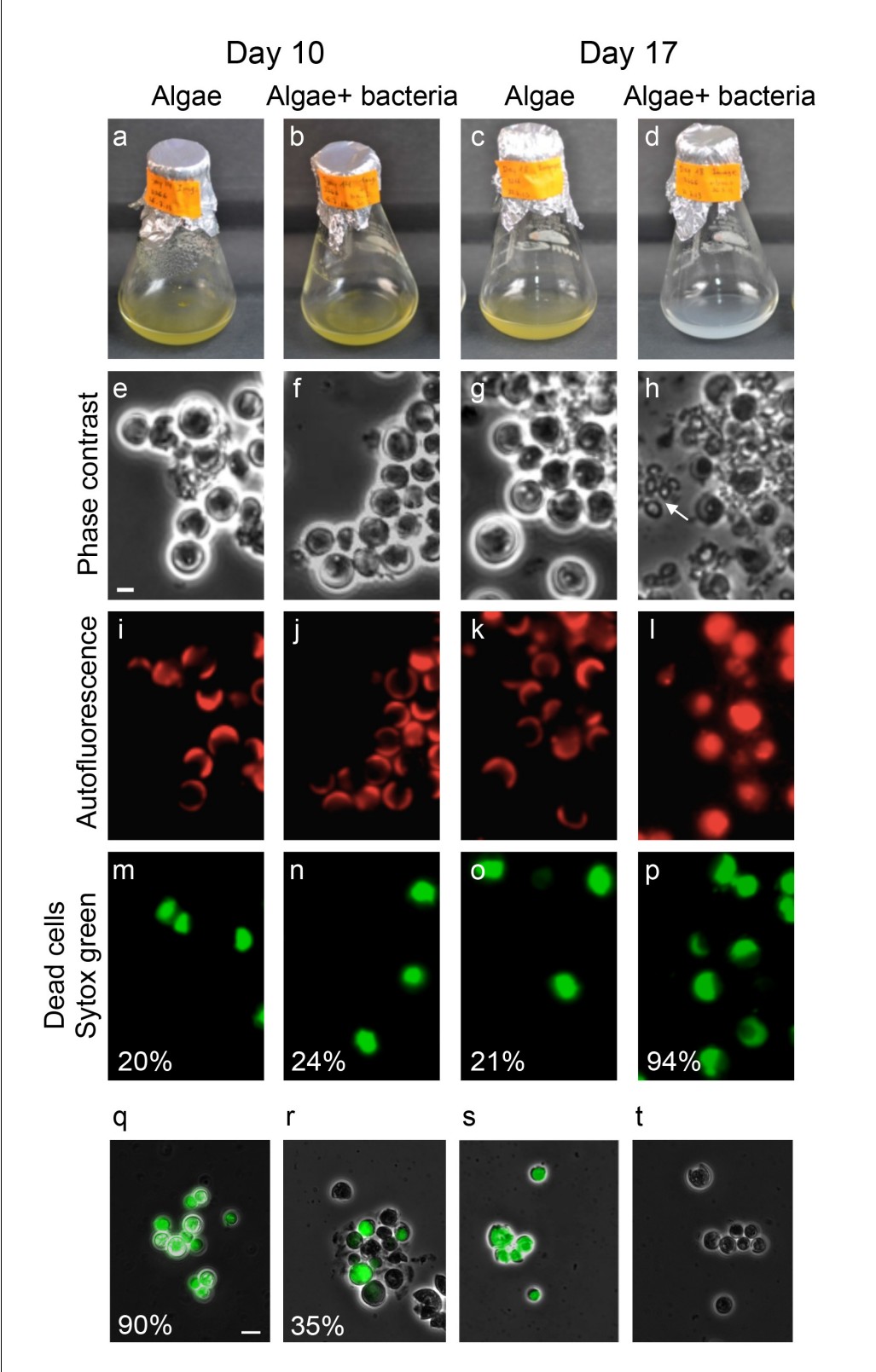

**Figure 6.** Bacteria induce a unique algal death in aging co-cultures. (a–d) Images of cultures demonstrating the change in the culture color over time. (e–h) Phase contrast microscopy images. Arrow points to shed coccoliths. (i–l) Fluorescent images of chlorophyll and accessory pigments autofluorescence. (m–p) Fluorescent images of dead cells stained with Sytox green. Percentages indicate the number of dead cells counted in each population. For each value n > 300 and the standard deviation between several analyzed fields was up to 20% of the indicated value. (q–t) Overlay of
*Figure 6 continued on next page*

*Figure 6 continued*

phase contrast microscopy images (grey) with fluorescent images of TUNEL assay (green) of cultures at day 20 (see Materials and methods). (**q**) Co-culture, (**r**) Axenic algal culture, (**s**) Positive control, cells were pretreated with DNase I, (**t**) Negative control, the terminal deoxynucleotidyl transferase enzyme (TdT) was replaced with distilled water. Percentages indicate the number of positively stained cells counted in each population. For each value n > 300 and the standard deviation between several analyzed fields was up to 25% of the indicated value. Scale bar corresponds to 1 μm in e–p and 4 μm in q–t.

that are directly involved in the metabolism of reactive oxygen species (ROS), such as glutathione (*Table 1*), was previously shown to be essential in viral infection of *E. huxleyi* (*Sheyn et al., 2016*). To further investigate if algae in co-culture experience a bacterially-induced process similar to PCD, algal cells from axenic cultures and co-cultures were tested for the presence of degraded DNA, a physiological hallmarks of PCD (*Gavrieli et al., 1992*). The fluorescent Terminal deoxynucleotidyl Transferase dUTP nick end-labeling assay (TUNEL) was previously established as a reliable method for in situ identification of DNA fragmentation as a PCD indicator in various phytoplankton species (*Johnson et al., 2014*; *Segovia et al., 2003*; *Berman-Frank et al., 2004*). Using the TUNEL assay (see Materials and methods) we could detect that 90% of algal cells in 20-day old co-cultures contain highly fragmented DNA in comparison to 35% in axenic algal cultures of the same age (*Figure 6q–t*).

To better understand how bacteria cause algal death in our model system, we explored the possible involvement of known algicidal compounds produced by *P. inhibens*. A previous study carried out in our laboratories reported on a novel group of compounds – roseobacticides – that are produced by *P. inhibens* and lyse *E. huxleyi* (*Seyedsayamdost et al., 2011*). We conducted several experiments to examine whether roseobacticides are the killing agents of algae in aging co-cultures. We successfully reproduced the findings of the previous study (*Figure 7*). However, our results indicate that while roseobacticides from *P. inhibens* indeed kill the non-calcifying algal strain CCMP372 used in the previous study, they do not kill the calcifying algal strain CCMP3266 used in the current study (*Figure 7a*). Importantly, in CCMP3266 cultures no bleaching was seen upon addition of roseobacticides. Moreover, there is no detectable production of roseobacticides in our co-culture conditions (*Figure 7b*). Thus, roseobacticides do not appear to play a role in algal death in the co-culture experimental system described here.

One small molecule released by bacteria that does appear to have a role in algal death is IAA. At concentrations between 1 and 100 μM IAA had a positive effect on algal growth, however 1000 μM proved to be harmful to algae (*Figure 4c*). The hormetic effect of IAA has been previously described; IAA promotes plant growth at low concentration while acting as a growth inhibitor at high concentrations (*Persello-Cartieaux et al., 2001*, *2003*; *Xie et al., 1996*). Similar effects were observed when increasing concentrations of IAA were added to a culture of diatoms (*Amin et al., 2015*). In addition, there are morphological similarities between algae that were killed by bacteria in co-culture and algae that were killed by high concentrations of IAA in pure culture (*Figure 4d*). These observations drove us to hypothesize that IAA can mediate both mutualistic and pathogenic interactions between bacteria and algae. Initially, bacterially-produced IAA promotes algal growth while later on it might be the driving force underlying algal death.

In various bacteria, attempts to perturb IAA synthesis were unsuccessful due to multiple redundant biosynthetic pathways (*Spaepen et al., 2007*). We examined several mutants of *P. inhibens* that carry mutations in various pathways of IAA synthesis (*Table 2* and *Figure 8a*). In agreement with previous reports, these mutants were still capable of producing IAA (*Table 2*) (*Spaepen et al., 2007*) and cause

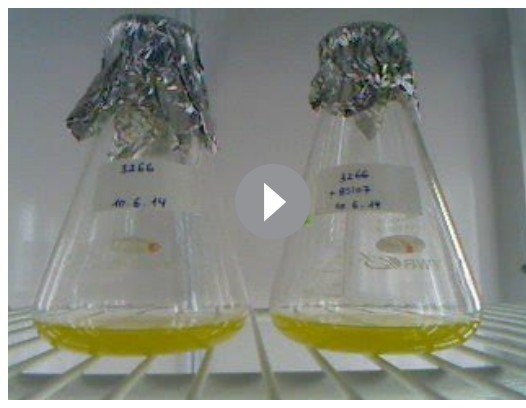

**Video 1.** Live imaging of the algal bleaching in co-culture.

**Table 1.** Expression data ratios between algae in co-culture and in axenic culture. Select genes involved in stress response and programmed cell death show up-regulation over time.

| Functional group | Gene annotation | Target transcript | Day 3 | Day 10 | Day 17 |
|---|---|---|---|---|---|
| Oxidative stress | putative L-ascorbate peroxidase | XM_005793355 | 1.36 | 3.19 | 6.81 |
| Programmed cell death | putative programmed cell death protein (PDCD2) | XM_005768970 | 1.31 | 2.13 | 5.29 |
| Metacaspases | putative metacaspase protein with Ca-binding EF hand domain | XM_005784588 | 1.31 | 3.30 | 4.54 |
| Metacaspases | putative metacaspase protein | XM_005763016 | 1.05 | 2.32 | 4.39 |
| Oxidative stress | ascorbate oxidase (AO) | XM_005775302 | 1.69 | 2.82 | 4.09 |
| Oxidative stress | putative glutathione-S-transferase | XM_005761417 | 1.31 | 2.20 | 4.07 |
| Metacaspases | putative metacaspase protein | XM_005791576 | 1.99 | 2.65 | 4.03 |
| Programmed cell death | putative death-specific protein with Ca binding EF hand domain | XM_005778875 | 1.48 | 2.40 | 3.88 |
| Programmed cell death | putative programmed cell death protein (PDCD2) | XM_005790372 | 1.21 | 2.12 | 3.84 |
| Metacaspases | putative metacaspase protein | XM_005773908 | 1.46 | 2.28 | 3.83 |
| Oxidative stress | glutathione synthetase (GSHS3) | XM_005760150 | 1.32 | 2.14 | 3.56 |
| Oxidative stress | putative L-ascorbate peroxidase | XM_005784352 | 1.17 | 2.19 | 3.33 |
| Programmed cell death | putative death-specific protein with Ca binding EF hand domain | XM_005773034 | 1.41 | 2.00 | 3.22 |
| Metacaspases | putative metacaspase protein | XM_005759676 | 1.17 | 1.90 | 3.06 |
| Oxidative stress | thioredoxin | XM_005761968 | 1.03 | 2.01 | 2.75 |

algal bleaching. Of note, a mutant in tryptophan biosynthesis (*Table 2*, transposon mutant 1630) was unable to grow in pure culture without addition of tryptophan (data not shown) but exhibited robust growth in co-culture with algae (*Figure 8b*) and was able to drive algal bleaching. These results further corroborate the presence of tryptophan in algal exudates.

To test whether high concentration of bacterially-produced IAA could cause algal death, we wanted to promote the biosynthesis of IAA by bacteria in co-culture. Because *P. inhibens* exhibited a 25-fold increase in yields of IAA upon utilization of exogenous tryptophan (*Figure 5a* and *Table 2*), we reasoned that an increase in IAA production should occur when a co-culture is supplemented with exogenous tryptophan. Our results indicate, that a co-culture that has been supplemented with 0.1 mM tryptophan undergoes accelerated death with bleaching occurring one week earlier than in non-treated co-cultures (*Figure 9a*). Moreover, examination under the microscope of co-cultures that have been treated with tryptophan revealed a remarkable change in bacterial behavior. In cultures supplemented with tryptophan, bacteria became hyper-colonizers and attached to algal cells in significantly higher rates (*Figure 9b*). Taken together, our observations indicate that tryptophan serves both as a precursor for IAA biosynthesis and as a cue capable of altering bacterial behavior towards their algal host.

The process in which algal biomass gradually increases to form a bloom, covering vast areas and then abruptly collapses, has attracted much attention from various disciplines. It is known that viral infection is key in the blooms' demise (*Bratbak et al., 1993*; *Lehahn et al., 2014*; *Vardi et al., 2012*). In our model system the presence of one species of bacteria resulted in the sharp demise of the algal population. While further studies are required in order to explore *E. huxleyi*-bacterial interactions in the ocean, it is possible that bacterial influences act in concert with viruses to drive the termination of the blooms. Environmental stresses such as nutrient depletion as well as viral infection have been shown to enhance PCD in *E. huxleyi* (*Bidle, 2015*, *2016*). Similarly, we have observed bacterially-induced PCD. While attached bacteria seem to be trapped on their dying host (*Figure 10*), their offspring get access to the nutrients from lysed algae and can then swim away and colonize a younger algal cell. Recently, Smirga and colleagues demonstrated experimentally how planktonic bacteria crowd around a lysing microalga, feeding off the released cellular content (*Smriga et al., 2016*). Our discovery of chemical crosstalk in the phycosphere and the bacterially-mediated algal death were made possible due to the co-culture experimental system that enables investigation over time.

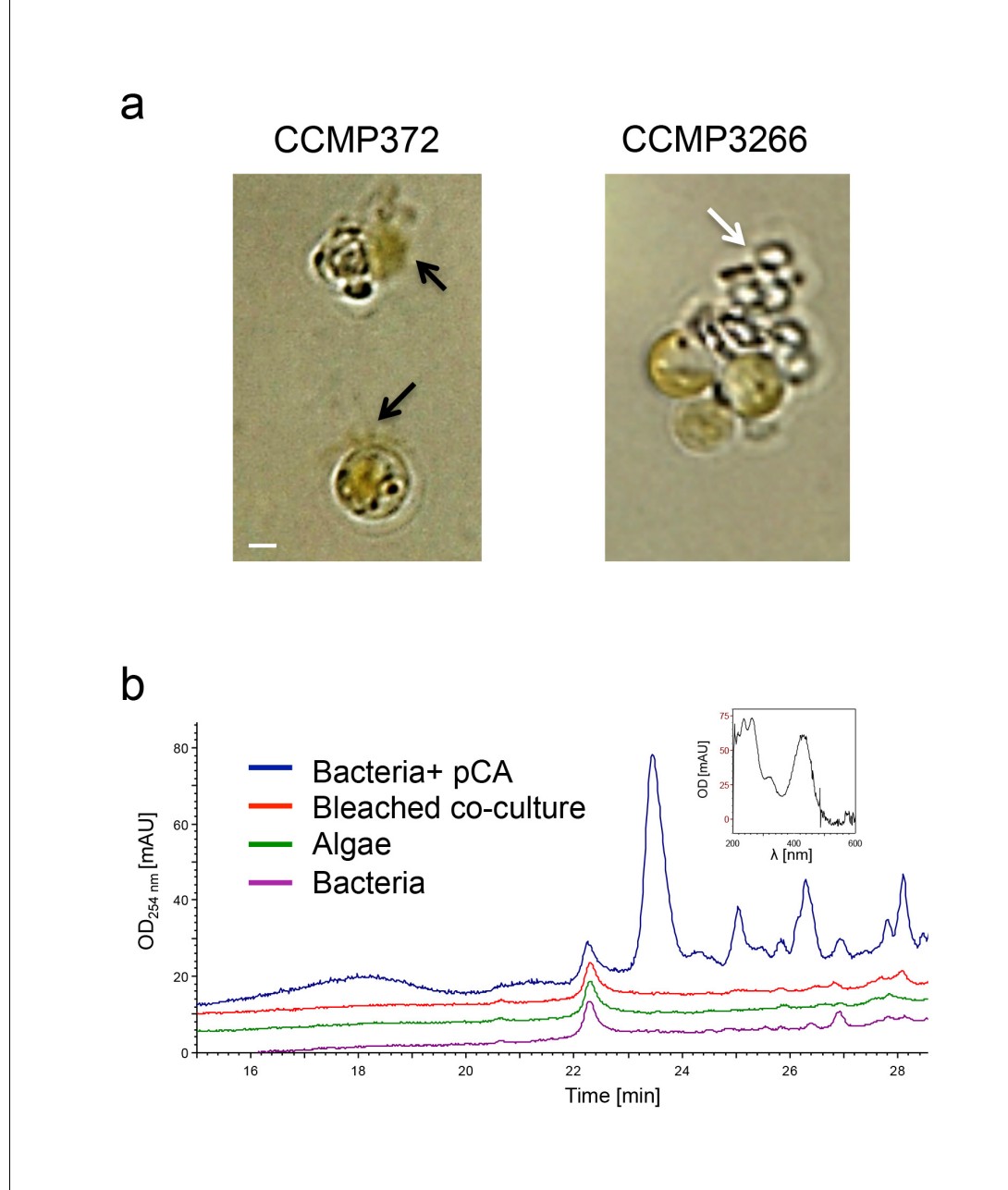

**Figure 7.** Roseobacticide-mediated algal lysis is strain specific. (**a**) Roseobacticides were introduced into 1 ml of algal cultures of *E. huxleyi* strains CCMP372 and CCMP3266. After 12 hr, cell lysis accompanied by chloroplast discharge was evident only in strain CCMP372 (black arrows). White arrow points to coccoliths in the image of the calcifying strain CCMP3266. Control cultures treated with equivalent volumes of solvent (methanol) or medium, did not exhibit any change (not shown). Scale bar corresponds to 1 µm. (**b**) Roseobacticides were extracted from various cultures (see Materials and methods). Roseobacticides were detected in bacterial culture grown in the presence of para-coumaric acid ('Bacteria + pCA', blue). Inset showing the characteristic absorbance peak of Roseobacticides at 430 nm (*Seyedsayamdost et al., 2011*). No Roseobacticides were detected in a bleached co-culture (red) or in control pure cultures of algae (green) or bacteria (purple) grown without addition of pCA.

In the current study tryptophan was identified as a central compound in the *E. huxleyi – P. inhibens* interaction. This led us to explore its abundance in *E. huxleyi* natural blooms, where it indeed was found. Previously, tryptophan was identified as a key metabolite in the interaction between a diatom and a Roseobacter (*Amin et al., 2015*). The use of both targeted and untargeted metabolomics in the lab can greatly aid in identifying metabolites with ecological relevance (*Fiore et al.,*

**Table 2.** *P. inhibens* mutant strains used in the current study.

| Strain | Genotype | Hypothetical function | IAA production in CNPS [nM] | IAA production in CNPS + trp [nM] |
|---|---|---|---|---|
| DSM 17395 | Wild type | | 0.4 | 10.8 |
| 3756 | PGA1_c11890:: Tn5 (KanR) | Indoleacetamide hydrolase | Yes | Yes |
| 3796 | PGA1_c11890:: Tn5 (KanR) | Indoleacetamide hydrolase | Yes | Yes |
| 1630 | PGA1_c16910:: Tn5 (KanR) | Indole-3-glycerol phosphate synthase | NA* | Yes |
| 1397 | PGA1_c23870:: Tn5 (KanR) | Nitrile hydratase subunit beta | Yes | Yes |
| 3422 | PGA1_c31390:: Tn5 (KanR) | Aromatic-L-amino-acid decarboxylase | Yes | Yes |

*NA - Not available since this mutant does not grow in these conditions without the addition of tryptophan

*2015*; *Johnson et al., 2016*; *Durham et al., 2015*; *Amin et al., 2015*). As revealed in the current study, when microorganisms are in close proximity, molecules can be produced and rapidly consumed and thus remain undetectable. Local and short-lived signals that are significant cues in the phycosphere would be undetectable in the environment but can be deciphered using model systems similar to the one we have described.

In our study, IAA was identified as a key component of the algal-bacterial chemical crosstalk. The bacterially-produced IAA initially increases algal yields. This growth enhancement is followed by an inevitable death. Clearly IAA exhibits a hormetic effect - it is beneficial in low concentrations and becomes harmful at higher doses. The hormetic nature of IAA has been previously demonstrated in various plant-bacteria interactions (*Persello-Cartieaux et al., 2001*, *2003*; *Xie et al., 1996*). In the association of the plant pathogen *Agrobacterium tumefaciens* with its hosts, IAA is central too (*Spaepen et al., 2007*; *Subramoni et al., 2014*). It is produced to enhance growth of plant tissue while creating tumors. Several similarities between *A. tumefaciens* and *P. inhibens* seem to exist; both attach through their pole to their host (*Heindl et al., 2014*; *Xu et al., 2013*), both express a polar polysaccharide involved in bacterial attachment (*Heindl et al., 2014*; *Xu et al., 2013*) and both utilize IAA to manipulate the growth of their host (*Subramoni et al., 2014*). In light of these resemblances, it is possible that additional aspects of the *E. huxleyi* – *P. inhibens* interaction will be similar to the mechanisms employed by *A. tumefaciens* and its interactions with host plants.

We have shown that IAA produced by bacteria enhances algal growth. How is algal growth promoted? One possibility is that algal growth augmentation is the result of changes in light harvesting efficiency. As reported by Falkowski and colleagues, changes in the capability of transducing light energy to chemical energy and eventually to biomass will result in changes in growth (*Falkowski et al., 1985*). Since all our cultures have been cultivated under the same light regime, changes in light harvesting capabilities would be the result of changes in levels of light harvesting pigments. However, chlorophyll a measurements revealed similar concentrations in algal cells over a period of 17 days, whether in the presence or absence of bacteria (*Table 3*). Only after bleaching, when the majority of the algal population was dead, a marked decrease in chlorophyll a concentration was evident in agreement with the macroscopic phenotype of bleaching (*Figure 6d*) and microscopic phenotype of chloroplast deformation (*Figure 6l*) that are seen in the co-culture at the same time. Thus, enhanced algal growth does not seem to be the result of more efficient light harvesting. The mechanisms underlying increased algal growth in response to IAA remain unknown.

IAA-induced growth in our system could represent the loss of growth control by *E. huxleyi*. In various microorganisms, regulatory circuits lead to growth cessation before resources have been completely exhausted. When such controls are bypassed, growth does not stop and higher yields are obtained. Two examples can be discussed in the context of elevated growth as a consequence of loss of growth control; *E. coli* mutants in *rpoS* grow to higher cell numbers than the wild type (*Vulic and Kolter, 2001*). While the wild type senses the near depletion of essential nutrients and prepares in advance by ceasing growth, the mutant lacks this regulatory ability (*Vulic and Kolter, 2001*). Another example of growth regulation by microorganisms is the ability of cells to sense their critical density, a process referred to as quorum sensing (QS). This process requires the release and

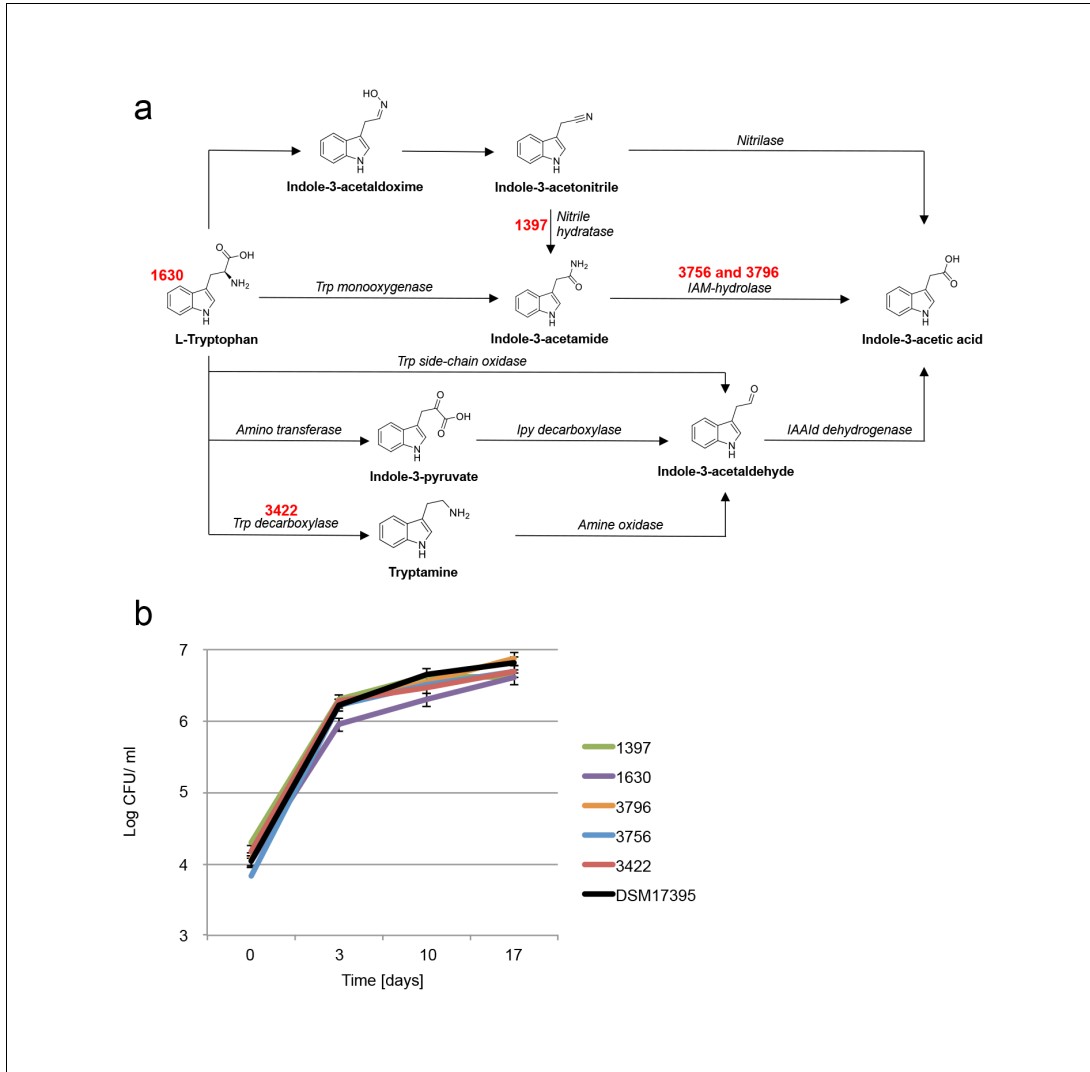

**Figure 8.** Examination of bacterial mutants. (**a**) Mapping the different mutants onto the various bacterial IAA biosynthetic pathways. Mutant strain numbers (see **Table 2**) are indicated in red next to the pathway that was mutated. Known bacterial enzymes appear in italic. Trp - tryptophan, IAM - indole-3-acetamide, Ipy - indole-3-pyruvate, IAAId - indole-3-acetaldehyde. Image modified from **Spaepen et al. (2007)**. (**b**) Mutants in various biosynthetic pathways of indole-3-acetic acid (IAA) production were grown in co-culture with algae over a period of 17 days. As can be seen, all mutants exhibited growth dynamics similar to the wild type DSM 17395 strain. Of note, mutant 1630 that is a tryptophan auxotroph (see **Table 2**) could not grow in pure culture in the absence of tryptophan but was able to grow in co-culture. Error bars represent the standard deviation of two biological replicates.

The following figure supplement is available for figure 8:

**Figure supplement 1.** Agarose gel of PCR amplifications confirming the transposon insertion site of select sequenced mutants in *P. inhibens*.

sensing of signal molecules that serve as reporters of the population density. The response to increased cell numbers is often a transition to decreased growth, which promotes survival of the cells under conditions of limited resources. In trypanosomes, perturbed ability to sense signal molecules results in failure to undergo growth arrest and leads to uncontrolled proliferatation (**Mony and Matthews, 2015**). Whether *E. huxleyi* possesses regulatory circuits to control growth cessation, and whether IAA can influence such regulation, is yet to be explored.

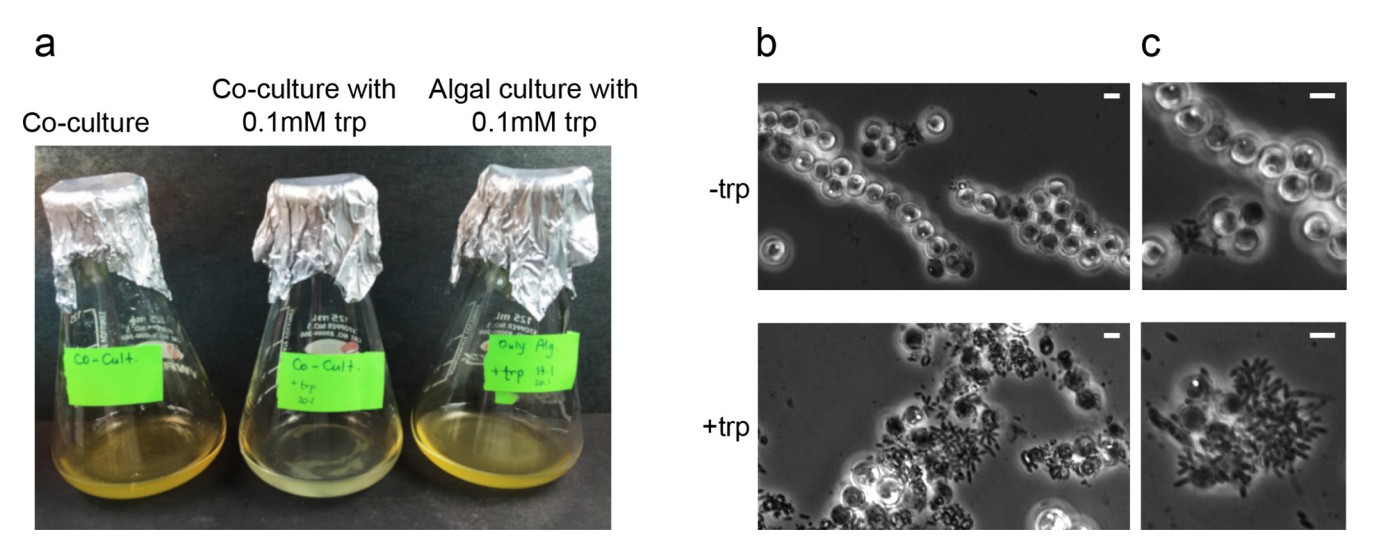

**Figure 9.** In bacteria, exogenous tryptophan serves as both a precursor and a cue. (a) Images of co-cultures and algal cultures at day 10. Upon inoculation of bacteria, 0.1 mM tryptophan (trp) was added. Seen is a co-culture that bleached a week earlier (middle). An algal culture treated with the same concentration of tryptophan did not display bleaching (right). (b–c) Phase contrast microscopy imaging of co-cultures at day 10 with (+trp) and without (−trp) addition of 0.1 mM tryptophan. Scale bar corresponds to 3 µm. Note that in the tryptophan treated co-culture (+trp) in (c) each algal cell has attached bacteria and the bacterial coverage is so dense that several covered algal cells cannot be seen.

Bacteria in our co-cultures were found attached solely to naked algal cells. We did not detect bacteria attached to calcified algae or shed coccoliths. What underlies the specific bacterial attachment to naked algal cells? Roseobacters posses the ability to chemotax (*Miller and Belas, 2006*; *Miller et al., 2004*), thus it is likely that *P. inhibens* swims in its planktonic stage specifically towards the naked cells. In turn, this might suggest that naked cells release increased levels of exudates. However, the question still remains whether the algal cells are naked irrespective of bacteria, or whether bacteria encourage coccoliths shedding thus increasing the amount of naked algal cells. In every population of calcified *E. huxleyi*, a sub-population of naked non-calcified cells is seen. It has been previously demonstrated that microzooplankton display higher growth rates when feeding on

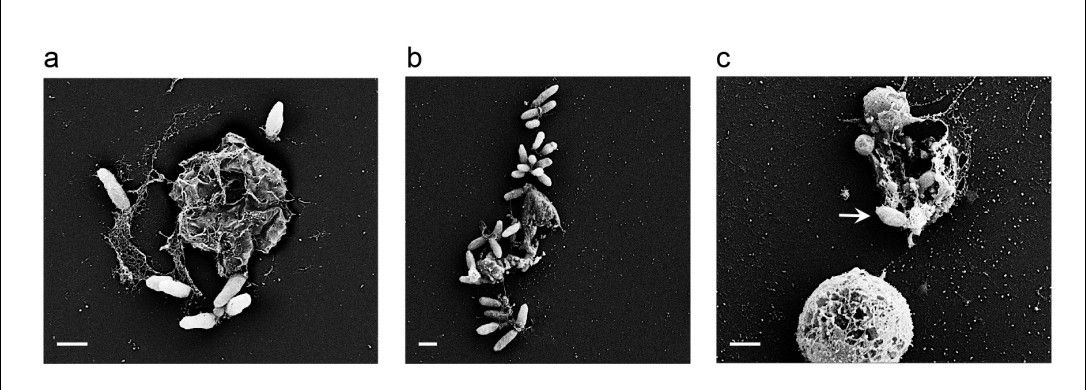

**Figure 10.** Bacteria attached to remains of dead algal cells. (a–c) SEM images of co-cultures at day 17 of incubation reveal bacteria that are attached to algal debris. The vast majority of imaged algal cells were intact (see for example the lower cell in panel c) indicating that algal debris were not generated as a result of sample preparation. Arrow pointing at a bacterial cell attached to algal remains that are adjacent to an intact algal cell. Scale bar corresponds to 1 µm.

**Table 3.** Chlorophyll a measurements in algal cultures. Chlorophyll a was measured spectroscopically in pure and co-cultures over time (see Materials and methods).

| Sample ID | Chlorophyll a per cell [pg] |
|---|---|
| Algae day 3 | $5.96 \times 10^{-3}$ |
| Co-culture day 3 | $6.18 \times 10^{-3}$ |
| Algae day 10 | $1.43 \times 10^{-2}$ |
| Co-culture day 10 | $1.67 \times 10^{-2}$ |
| Algae day 17 | $1.50 \times 10^{-2}$ |
| Co-culture day 17 | $6.55 \times 10^{-3}$ |

naked *E. huxleyi* cells in comparison to calcified cells (*Harvey et al., 2015*). Thus, bacteria might initially target a pre-existing algal sub-population. Reversible modes of attachment were shown to take place when bacteria such as *A. tumefaciens* are assessing whether or not to attach to a substrate in a new environment (*Heindl et al., 2014*). As *P. inhibens* shares several similarities with this bacterium regarding their mode of attachment (*Segev et al., 2015*), it is possible that *P. inhibens* evaluates the algal cells through reversible interactions. In this regard, the physical properties of a naked noncalcified algal cell might be more suitable for bacterial attachment. Similar influences might promote the permanent attachment that persists even after the algal cell has died (*Figure 10*).

In this study we have established a well-defined microbial model system and carried out a detailed characterization of micro-scale interactions over time. The discovery of a dynamic microbial interaction and the mechanisms underlying it was made possible due to the use of a simplified experimental system for coccolithophore-bacteria interactions. Exploration of similar robust model systems could further reveal chemical and molecular details that might be studied in the ocean. Attempts to link laboratory findings and environmental studies will both widen and deepen our understanding of microbial interactions, their ecophysiology and the extent to which these interactions influence the marine environment.

## Materials and methods

### Strains and general growth conditions

The bacterial strain of *Phaeobacter inhibens* was DSM 17395 purchased from the German collection of microorganisms and cell cultures (DSMZ, Braunschweig, Germany). Bacterial cultures were grown in liquid 1/2YTSS medium containing 2 g yeast extract (BD, NJ, USA), 1.25 g tryptone (Sigma-Aldrich, MO, USA) and 20 g sea salt (Sigma-Aldrich) per liter. Cultures were incubated at 30°C shaking at 130 rpm.

The axenic algal strain of *Emiliania huxleyi* was CCMP3266 purchased from the National Center for Marine Algae and Microbiota (Bigelow Laboratory for Ocean Sciences, Maine, USA). Algae were grown in L1 medium according to (*Guillard and Hargraves, 1993*), with the exception that $Na_2SiO_3$ was omitted following the cultivation recommendations for this algal strain, and the medium was referred to as L1-Si. Algae were grown in standing cultures in a Percival chamber (Percival Scientific, IA, USA) at 18°C under a light/dark cycle of 12/12 hr. Illumination intensity during the light period was 150 µmoles/m$^2$/s. Cultures were maintained axenic using a mixture of penicillin and streptomycin at a final concentration of 0.1 mg/ml and 0.05 mg/ml respectively. The antibiotic-treated cultures were used to inoculate antibiotic-free cultures. At least two passages through antibiotic-free medium were carried out prior to inoculation of the experimental cultures. Absence of bacteria in axenic algal cultures was monitored periodically both by plating on 1/2YTSS plates and under the microscope.

Co-cultures of *E. huxleyi* and *P. inhibens* were cultured as follows: 14-day-old *E. huxleyi* cultures were inoculated into L1-Si medium at a 1:100 dilution and incubated as described above. After four days of algal growth, a colony of *P. inhibens* was resuspended in 1 ml L1-Si and 100 µl were added to 10 ml of algal culture. The co-cultures were incubated in a Percival chamber under the conditions described above for algal cultures.

### Monitoring bacterial growth in co-cultures

In order to evaluate bacterial growth in co-cultures, samples were taken from co-cultures at different time points, as indicated. Samples were serially diluted and plated on 1/2YTSS plates. These plates

facilitate only bacterial growth, and thus counts of colony forming units (CFU) were indicative of bacterial numbers in the co-culture. In order to eliminate rosettes and obtain CFU from single cells, we attempted sonication prior to plating, however we were unable to eliminate rosettes and sonication did not alter the CFUs that were obtained. Thus, samples were not sonicated and CFU may include rosettes as well as individual bacteria.

## Scanning electron microscopy

Samples of 10 µl were placed on 1 cm$^2$ cover slips coated with poly-L-lysine and kept in a humid environment at room temperature for 1 hr. Samples were then submerged sequentially for 10 min in 5 ethanol solutions of increasing concentrations from 30% to 100% absolute ethanol. Following critical point drying the samples were sputter coated with an Au/Pd alloy. Images were obtained with a Zeiss Supra55 Field Emission scanning electron microscope.

## Light microscopy

Fluorescence and phase contrast images were obtained using a Nikon TE-2000U inverted microscope equipped with a 100× Plan Apo NA 1.4 objective lens. All samples were spotted on thin 1% agarose pads for visualization at room temperature. Images were acquired using a cooled Hamamatsu CCD camera controlled with MetaMorph seven software (Molecular Devices, CA, USA). Algal autofluorescence was visualized using a Cy5 long-pass filter (Chroma #41024). Alexa Fluor 488 conjugated WGA, Sytox green and TUNEL signals (see below) were captured using a narrow band eGFP filter (Chroma #41020). Phase contrast images where green chlorophyll is shown (*Figure 7*) were obtained with a Zeiss Axioskop two plus microscope equipped with a Zeiss Axiocam MRc camera and using a Zeiss plan-Apochromat 63x lens with a 1.4 NA. Images were analyzed using the MetaMorph seven software. Images were processed identically for compared image sets.

## Fluorescent stains

### Bacterial polar polysaccharide

Samples were incubated in the dark with Alexa Fluor 488 conjugated WGA (Life Technologies, CA, USA) for 30 min at room temperature. The stain final concentration was 5 µg/ml. Samples were rinsed twice in PBS prior to visualization.

### Dead cell staining

To stain dead cells for visualization under the microscope, cultures were incubated with Sytox green (Life Technologies). For flow cytometry, cells were stained with Sytox blue (Life Technologies). For both stains, cells were incubated at a final concentration of 5 µM. Incubation was carried out in the dark for at least 30 min at room temperature.

### Terminal deoxynucleotidyl transferase-dUTP nick end labeling (TUNEL) assay

DNA fragmentation was detected in situ using TUNEL labeling as previously described (*Segovia et al., 2003*), with slight modifications. Briefly, cells were fixed with 0.1% glutaraldehyde and 4% paraformaldehyde (w/v). Cells were permeabilized with 0.25% Triton X-100 (Sigma-Aldrich) in 1X PBS and labeled following the manufacturer protocol (Click-iT Plus TUNEL assay, Alexa Fluor 488, Molecular Probes, CA, USA). Following labeling, cells were resuspended in 1X PBS and visualized as specified above. Positive controls were generated by pretreating cells with 15 U ml$^{-1}$ DNase I (Thermo Fischer Scientific, MA, USA). These controls yielded a very high fluorescent signal and thus were scaled differently to avoid image saturation. For negative controls, the enzyme terminal deoxynucleotidyl transferase was replaced with distilled water. Quantification of positively stained cells in the population was conducted as follows; all fluorescent images aside for the positive controls were scaled identically. Based on the negative controls, a fluorescent threshold was set to eliminate any background signal. Fluorescent images were overlayed with phase contrast images and cells with and without a fluorescent signal were manually counted.

## Quantification of bacteria per algal cell

Phase contrast images of co-cultures at the specified ages were obtained and bacterial cells were manually counted using the MetaMorph seven software. In the case of algal clumps, the total number of attached bacteria was divided by the number of algae in the clump. Given the three dimensionality of cells and clumps, bacterial numbers are most likely underestimated, as bacteria that attach under the observed specimen cannot be seen. For all time points n > 300.

## Bacterial growth upon external addition of essential nutrients

To assess the identity of essential nutrients provided by algae to support bacterial growth, bacteria were grown in L1-Si medium (in which they cannot grow) and essential nutrients were externally added. All nutrients were added in forms that were previously shown to be utilized by *P. inhibens* (*Zech et al., 2009*); glucose 5.5 mM, $Na_2SO_4$ 33 mM, $NH_4Cl$ 5 mM, $KH_2PO_4$ 2 mM, all purchased from Sigma-Aldrich. The sulfur source was replaced with 30 µM or 6 mM dimethylsulfoniopropionate (DMSP, Research Plus, NJ, USA) where noted.

## Flow cytometry

Cultures were grown as previously described. Prior to analysis, 250 µl of each culture were supplemented with the viability dye Sytox blue as mentioned above (Life Technologies) and 50 µl of Count-Bright fluorescent beads (Life Technologies). The fluorescent beads were used to calculate cell numbers according to the manufacturer's instructions. Samples were analyzed on an 18-channel FACSAria SORP flow cytometer (BD Biosciences, MA, USA). For each sample 100,000 events were recorded. Data analysis was carried out using the DIVA software version 6.3.1 (BD Biosciences). In order to validate the accurate interpretation of particles in the analyzed samples, select samples were additionally analyzed using the ImageStream 100 multispectral imaging flow cytometry instrument (Amnis, WA, USA) equipped with 405, 488 and 658 nm laser sources with variable laser power, and a brightfield source. Prior to analysis, samples were transferred to 500 µl siliconized microcentrifuge tubes (Sigma-Aldrich). Data of boiled and/or stained control algae were collected and used to compensate fluorescence spectral overlap between the different fluorescent channels, and set the optimal laser power to avoid saturation of the camera. Samples were gated for single cell population using the area and aspect ratio features as previously described (*Ponomarev et al., 2011*). Analysis was carried out while minimizing the 658 nm laser power to decrease bleed through of chlorophyll and accessory pigments autofluorescence. Data files contained 10,000–20,000 cells and were analyzed using the Image Data Exploration and Analysis Software (IDEAS) (Amnis).

## DMSP analysis

### Sample preparation

Cultures of algae in the presence or absence of bacteria were grown and at the specified time points, 10 ml of each culture was filtered through a 0.2 µm syringe filter (Pall, NY, USA). Filtrates were stored at −20°C for no longer than one week prior to analysis.

### Liquid chromatography coupled with mass spectrometry analysis (LC-MS)

For initial identification of DMSP in the samples, a Bruker maXis impact Q-TOF LC-MS (Bruker, MA, USA) with an Agilent 1290 HPLC was used (Agilent, CA, USA). The column used was a Jupiter C4 column (5 µm, 4.6 mm × 150 mm) (Phenomenex, CA, USA). The solvent system consisted of 4 mM ammonium formate (AmFo) with 0.1% formic acid in water (A) and 0.1% formic acid in acetonitrile (B). Samples were eluted using the following mobile phase composition gradient: 0–7 min from 0% to 90% B; 7–15 min at 90% B. The analyses were performed at a flow rate of 0.4 ml/min, and DMSP was detected by HR-MS in positive ionization mode; retention time and MS/MS fragments were compared with the standard compound. The standard for DMSP was purchased from Research Plus.

Quantitative analysis of DMSP was carried out using an Agilent 1200 Series analytical system LC-MS equipped with a photodiode array (PDA) detector combined with a 6130 Series ESI mass spectrometer. Formic acid (>99.5%) was obtained from Sigma, and LC-MS-grade acetonitrile ($CH_3CN$) and water were purchased from Fisher Scientific. Filtered samples were analyzed using a Phenomenex phenyl hexyl analytical column (5 µm, 4.6 mm × 100 mm). The mobile phase was a gradient program of mixtures of 0.1% formic acid in water (A) and 0.1% formic acid in acetonitrile (B), the

method was as follows: 0–40 min from 2% to 98% B; 40–45 min at 98% B; followed by a change to to 2% B at 45.5 min, and then isocratic conditions with 2% B to 55 min (total 55 min). The flow rate was set at 0.5 ml/min, and the injection volume was 10 μl. Calibration curves and linear regression equations were generated for the external standard. Quantification of DMSP was based on the obtained peak area and calculated as equivalents of the standard.

## IAA detection in bacterial cultures
### Sample preparation
*P. inhibens* bacteria were cultured in 50 ml minimal medium containing L1-Si supplemented with 5.5 mM glucose, 33 mM $NaSO_4$, 2 mM $KH_2PO_4$ and 5 mM $NH_4Cl$. When indicated, cultures were supplemented with 10 mg tryptophan or 5 mg $^{15}N/^{13}C$ labeled tryptophan at the beginning of the culture period. Cultures were incubated four days at 30°C shaking at 130 rpm. Cultures were centrifuged in 50 ml Falcon tubes at 8000 rpm for 10 min at 4°C. The supernatant was transferred to a clean 50 ml Falcon tube and acidified with formic acid to pH = 3. Acidified supernatant were loaded onto Oasis HLB cartridges (Waters, MA, USA) and eluted with methanol. Eluted samples were dried *in vacuo* and resuspended in 200 μl methanol. Samples were filtered prior to high resolution LC-MS analysis.

### High resolution LC-MS analysis
LC-MS data were acquired using an Agilent 6530 ESI-QTOF mass spectrometer with an Agilent 1290 UHPLC system. Using a flow rate of 0.3 ml/min on an RP C18 column (Phenomenex Kinetex 2.6 μm, 2.1 mm × 100 mm), acetonitrile/$H_2O$ containing 0.1% formic acid (10%/90%) was held for 1 min, followed by a linear gradient of acetonitrile and $H_2O$ (containing 0.1% formic acid) from (10%/90%) to (100%/0%) in 11 min, and finally, held for 2 min at acetonitrile/$H_2O$ (100%/0%). Full scan mass spectra (m/z 100–1700) were measured in positive ESI mode. The mass spectrometer was operated with the following parameters: nebulizer pressure, 2.76 bar; dry gas flow, 10.0 L/min; dry gas temperature, 325°C; capillary, 3.5 kV; scan rate, 1 Hz. Two internal calibration compounds, purine and hexakis (1 hr, 1 hr, 3H-tetrafluoropropoxy) phosphazine, were used at low concentration throughout the acquisition. Targeted MS/MS data were acquired using two collision energies: 10 and 20 eV. IAA standard was purchased from Research Products International Corp (IL, USA).

## Growth of *E. coli* tryptophan auxotroph in algal supernatant
*E. coli* strain CGSC#6666 was obtained from the Coli Genetic Stock Center (Yale University, New Haven, CT). The strain was grown overnight in M9 medium supplemented with 0.4% glucose and 10 mM tryptophan. In the following morning, cells were centrifuged, washed twice with M9 + 0.4% glucose and diluted to $OD_{600}$ = 0.05 into a 96-well plate. Each well contained 100 μl culture and 100 μl L1-Si or algal supernatant where specified. Treatments included no addition of tryptophan, addition of tryptophan (30 or 300 μM), or addition of algal supernatant filtered from a 14-day old algal culture. Each treatment was tested in four replicates. The plate was incubated at 30°C in a SPECTRA max M2 plate reader (Molecular Devices). Every 30 min the plate was shaken for 5 s and $OD_{600}$ was measured, over a course of 16 hr.

## Gene expression data
### Sample preparation and cell lysis
25 ml of each culture at the specified age were supplemented with 25 ml of RNAlater and incubated 30 min on ice. Samples were centrifuged 45 min at 8000 rpm at 4°C. The pellet was resuspended in 10 ml of RNase-free water (UltraPure, Invitrogen, CA, USA) and centrifuged 10 min at 8000 rpm at 4°C. Pellet was resuspended in 500 μl RNase-free water (UltraPure, Invitrogen) and centrifuged 2 min at 13,000 rpm at 4°C. Supernatant was discarded and pellet was quickly frozen in a dry ice + ethanol bath for 10 min. Frozen pellet was thawed on ice for 30 min and then resuspended in 500 μl RLT buffer (Qiagen, Germany) containing 1:100 β-mercaptoethanol (Sigma-Aldrich). Samples were vortexed and placed on ice for 10 min. Samples were then transferred to tubes containing 0.1 and 0.5 mm glass beads and subjected to four cycles of 1 min at the highest setting in a Mini beadbeater (BioSpec Products, OK, USA), with 30 s cooling on ice between cycles. Samples were then centrifuged 5 min at 8000 rpm at 4°C, supernatants were collected and stored at −80°C until analysis.

## Gene expression analysis

Lysates were diluted five-fold for hybridization. Isolation and quantification of RNA by NanoString nCounter was done according to the manufacturer's instructions (NanoString Technologies, WA, USA). Experiments were conducted in duplicates and gene expression data from all experiments was normalized as one dataset. Normalization was performed manually following the nSolver software recommendations in the following way; for each individual experiment the geometric mean was determined for select house keeping genes (HKGs). HKGs included alpha-tubulin (XM_005762286), beta tubulin (XM_005781549), 60 s ribosomal protein L13 (XM_005781721) and putative ribosomal protein L30 (XM_005790519).

Each geometric mean was divided by the average value of all geometric means and the resulting value was used as the normalization factor for all values in each individual experiment. To determine the threshold for background expression, the mean and standard deviation was calculated for all negative controls (controls that were included by the manufacturer). Background expression was determined as mean+(2*standard deviation), and expression data lower than this value were discarded.

Abundances of each mRNA were averaged between two biological replicates (see *Supplementary file 3*) and gene expression values are presented as ratio of expression in co-culture divided by expression in axenic algal culture.

## Generation of *P. inhibens* mutants

Preparation of electrocompetent cells from *P. inhibens* DSM 17395 was conducted as previously described (*Petersen et al., 2011*). Transposon mutagenesis in *P. inhibens* DSM 17395 was performed with the EZ-Tn5 <R6Kγori/KAN-2>Tnp Transposome kit (Epicentre, Illumina, CA, USA). Individual transposon mutants were cultured in MB medium (BD) with 120 μg ml$^{-1}$ kanamycin. Total DNA was isolated with the DNeasy Blood and Tissue Kit (Qiagen) and the insertion sites of 4000 transposon mutants were determined via arbitrary PCR as previously described (*O'Toole and Kolter, 1998*).

Select mutants (*Table 2*) were streaked out three subsequent times on MB plates containing 120 μg ml$^{-1}$ kanamycin in order to eliminate wild type cells that could survive in rosette structures. Subsequently, the precise integration site of each mutant strain was validated via PCR amplification with specific primers (*Supplementary file 2* and *Figure 8—figure supplement 1*) followed by sequencing.

## Roseobacticides extraction and detection

500 ml Erlenmeyer flasks containing 50 ml of 1/2YTSS medium and 1 mM para-coumaric acid (Sigma-Aldrich) were inoculated with 0.5 ml of an overnight culture and incubated at 130 rpm at 30°C for three days. Then, cultures were extracted once with an equal volume of ethyl acetate, dried *in vacuo*, resuspended in methanol and analyzed by HPLC. HPLC analysis was performed on a Beckman Coulter System Gold HPLC equipped with a diode array detector using an analytical Phenomenex C18 column (5 μm, 4.6 mm x 100 mm). Flow rate was 0.7 ml/min with a gradient of 10% acetonitrile in water to 100% acetonitrile over 25 min. Roseobacticides were identified according to their characteristic absorbance peak at 430 nm (*Seyedsayamdost et al., 2011*).

## Roseobacticides lysis assay

Cultures of *E. huxleyi* strains CCMP3266 and CCMP372 were grown seven days in L1-Si medium and then diluted with fresh medium 1:1 (v/v). Cultures were placed in 1 ml aliquots in a 48 well plate. Each well was supplemented with 10 μl of roseobacticide extract (as described above), methanol or medium. Samples were visualized under the microscope 12 and 24 hr after treatment.

## Metagenomic analysis

### Sampling area

Two *E. huxleyi* blooms were detected in the Gulf of Maine during the summer of 2015 based on acid-labile optical backscattering (*Balch et al., 1999*). Elevated calcite backscattering was detected in the Eastern edge of Jordan Basin on July 23rd and on the Western edge of Jordan Basin on September 17th. Two individual samples of 1 liter were collected from each bloom using a sampling

system described in Balch et al. (*Balch et al., 2012*). Samples were stored in sterile glass bottles and refrigerated in darkness for 48 hr until sample processing.

## Sample preparation, DNA extraction and sequencing

Whole Metagenomic shotgun sequencing was performed on the four samples described above. Samples were filtered through a 0.2 μm aPES membrane (Thermo Fisher Scientific, MA, USA). Filters containing biomass were frozen at −80°C. Prior to DNA extraction, filters were placed in a 50 ml falcon tube filled with filtered seawater and shaken vigorously. Tubes were then centrifuged at 8000 rpm at 4°C for 30 min. Supernatant was discarded and DNA was extracted using the ZR soil microbe DNA MiniPrep (Zymo research, CA, USA). DNA quality validation, library preparation and MiSeq sequencing were performed at the Biopolymers Facility at Harvard Medical School, Boston, MA. Paired end sequencing was conducted with 150 bp read from each end.

## Bioinformatic analysis

Whole metagenomic shotgun sequencing resulted in an average of 2,232,320 reads per sample with a standard deviation of 235,974 reads. Quality of reads was validated using FastQC version 0.11.4. Metagenomic reads were mapped using a k-mer based approach onto reference genomes retrieved from the RefSeq bacteria collection using Kraken version 0.10.5-beta (*Wood and Salzberg, 2014*). Approximately 35% of sequences were annotated as bacteria. 62% of bacterial sequences were assigned to the genus *Alteromonas* that are widespread non-specific marine bacteria often found attached or as free-living forms (*Acinas et al., 1999*). To better assess percentage of specific *E. huxleyi* associated bacteria, *Alteromonas* sequences were omitted. Consequently, percentage of the Rhodobacteraceae family, which was 2% of total bacterial sequences, was assessed as 6% in the absence of *Alteromonas*. Hierarchical visualization of data was performed using the Krona software (*Ondov et al., 2011*).

## Tryptophan measurement of algal bloom samples

Four samples of 1 liter were collected in two *E. huxleyi* blooms (see section above for details about the sampling area). Samples were filtered through a 0.2 μm aPES membrane (Thermo Fisher Scientific) and filtrates were collected in sterile bottles and stored at 4°C in the dark. Filtrates were acidified with formic acid to pH = 3. Acidified samples were loaded onto Oasis HLB cartridges (Waters) and eluted with methanol. Eluted samples were dried *in vacuo* and resuspended in 200 μl methanol. Samples were filtered prior to high resolution LC-MS analysis, which was preformed as described above.

## Chlorophyll a measurements

For Chlorophyll extraction, at the specified time points 5 ml of cultures were collected on ice and centrifuged for 20 min at 10,000 rpm at 4°C. Supernatant was discarded and pellets were resuspended in 1 ml of 90% acetone. Pellets were disturbed using vigorous pipettation and vortexing for 2 min. Samples were kept overnight at 4°C. Prior to Chlorophyll a measurements, samples were filtered through a 0.2 μm PVDF membrane (Pall) directly into cuvettes. Filtrate absorbance was measured on a Beckman DU640 spectrophotometer (Beckman Coulter, CA, USA) at the following wavelengths: 750, 663, 645 and 630 nm. Absorbance at 750 nm was subtracted from all other values to correct for turbidity. Chlorophyll a concentration was calculated using the following equation:

Chlorophyll a [μg/L] = {[11.64(Abs663) − 2.16(Abs645) + 0.10(Abs630)] $E$} / $V$($L$)

Where $E$ = the volume of acetone solution used for extraction (ml)

$V$ = the volume of sample filtered (L)

$L$ = the cuvette path length (cm)

The obtained values were divided by cell numbers determined using flow cytometry as described above.

## Acknowledgements

We are grateful to all the members of the Kolter lab for valuable discussions and assistance. We are thankful to Dr. William Balch and David Drapeau (Bigelow Laboratory for Ocean Sciences, East

Boothbay, ME) for their invaluable assistance in obtaining environmental samples. We thank Dr. Mor Grinstein and Dr. Jenna Galloway (Harvard Stem Cell Institute, Massachusetts General Hospital and Harvard Medical School, Boston, MA) for their generous help with the TUNEL assay. We thank Keith Ketterer from the Information Technology department at Harvard Medical School for help with video acquisition. We thank the Harvard Center for Nanoscale Systems for use of its imaging facility. This study was supported by fellowships from the European Molecular Biology Organization and from the Human Frontier Science Program granted to ES, DFG Transregio TRR-51 Roseobacter granted to JP, PCMM grant and NIH grant S10 RR023459 to NB., NIH grant GM086258 to JC and NIH grants GM58213 and GM82137 to RK.

## Additional information

### Competing interests

JC: Reviewing editor, *eLife*. The other authors declare that no competing interests exist.

### Funding

| Funder | Grant reference number | Author |
|---|---|---|
| European Molecular Biology Organization | LTF 649-2012 | Einat Segev |
| Human Frontier Science Program | LT000061/2013-L | Einat Segev |
| Deutsche Forschungsgemeinschaft | Transregio TRR-51 Roseobacter | Jörn Petersen |
| Program in Cellular and Molecular Medicine, Boston Children's Hospital | | Natasha Barteneva |
| National Institutes of Health | RR023459 | Natasha Barteneva |
| National Institutes of Health | GM086258 | Jon Clardy |
| National Institutes of Health | GM58213 | Roberto Kolter |
| National Institutes of Health | GM82137 | Roberto Kolter |

The funders had no role in study design, data collection and interpretation, or the decision to submit the work for publication.

### Author contributions

ES, Conception and design, Acquisition of data, Analysis and interpretation of data, Drafting or revising the article; TPW, KHK, NB, JNP, Acquisition of data, Analysis and interpretation of data, Drafting or revising the article; JP, Acquisition of data, Analysis and interpretation of data, Drafting or revising the article, Contributed unpublished essential data or reagents; CE, LC, Acquisition of data, Drafting or revising the article, Contributed unpublished essential data or reagents; HV, JC, RK, Conception and design, Analysis and interpretation of data, Drafting or revising the article

### Author ORCIDs

Einat Segev, http://orcid.org/0000-0002-2266-1219
Ki Hyun Kim, http://orcid.org/0000-0002-5285-9138
Jon Clardy, http://orcid.org/0000-0003-0213-8356
Roberto Kolter, http://orcid.org/0000-0001-9548-1481

## Additional files

### Supplementary files

• Supplementary file 1. Expression data ratios between algae in co-culture and in axenic culture.

• Supplementary file 2. Primer sequences for validation of transposon insertion sites.

• Supplementary file 3. Gene expression data.

### Major datasets

The following dataset was generated:

| Author(s) | Year | Dataset title | Dataset URL | Database, license, and accessibility information |
|---|---|---|---|---|
| Segev E, Wyche TP, Kim KH, Petersen J, Ellebrandt C, Vlamakis H, Barteneva N, Paulson JN, Chai L, Clardy J, Kolter R | 2016 | Dynamic Metabolic Exchange Governs a Marine Algal-Bacterial Interaction | https://www.ncbi.nlm.nih.gov/sra/?term=SRP075256 | Publicly available at the NCBI Short Read Archive (accession no: SRP075256) |

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
