## [Decision Letter]

Thank you for submitting your article "Dynamic metabolic exchange governs a marine algal-bacterial interaction" for consideration by *eLife*. Your article has been favorably evaluated by Detlef Weigel (Senior Editor) and three reviewers, one of whom, Paul G Falkowski (Reviewer #1), is a member of our Board of Reviewing Editors. The following individual involved in review of your submission has agreed to reveal their identity:; Kay Bidle (Reviewer #3).

The reviewers have discussed the reviews with one another and the Reviewing Editor has drafted this decision to help you prepare a revised submission.

Summary:

After careful examination of your response to our request, we will reconsider this paper for possible publication in *eLife*. Please understand that the paper will be re-evaluated by the three original reviewers. To that end, I strongly suggest that the "global" importance of the findings be toned down – as the evidence from field studies about the "global" importance is lacking. That aspect of the paper was particularly irritating to one of the reviewers.

*Reviewer #1:*

This is a very thorough paper with many results indicating that a specific bacterium can stimulate growth of a coccolithophore and subsequently induce an apoptotic cascade. The key finding is that the signaling molecule is IAA = which is curious. IAA receptors in the algal cell clearly can enhance growth – but I don't understand why that happens. If the cells actually grew faster at the same irradiance, then the enhancement should alter the quantum yield for growth – which the authors do not report. One key parameter that appears to be missing to make that calculation is the optical absorption cross section normalized to chlorophyll (so-called a*). If the algal cell reduced the production of chlorophyll at a particular irradiance, the a* value would be higher (less self-absorption) and the cell could actually use light more efficiently. After longer exposure to IAA (or the association), it appears that plastids are becoming disrupted. If the authors have chlorophyll/cell values, the a* can potentially be calculated – and that would help give some rationale to the stimulation of growth.

*Reviewer #2:*

There is some solid research here but it is spun into a story that is not fully supported by the data and, more importantly, much of the story was told a year ago in a paper in Nature. The new findings in the manuscript are the investigations of phenotypes of the phytoplankton and bacteria in response to IAA and tryptophan, but the concentrations needed to see the phenotypes are extremely high. This manuscript can be rewritten and shortened to highlight results that add to existing knowledge. But unfortunately, the study still will not address underlying mechanisms for phenotypic changes and will not constitute an advance sufficient for publication in *eLife*.

1) The discovery that bacterially-produced indole-3-acetic acid serves as a growth stimulant to marine phytoplankton and that tryptophan release by the phytoplankton fuels IAA release has already been discovered and published (Amin et al., Nature, 2015). This manuscript provides evidence that another bacterium in the same taxonomic group (Roseobacter) has a similar relationship with a phytoplankton in a different group (coccolithophore), and thus adds evidence to an earlier finding. However, it does not represent a new discovery.

2) Introduction, last paragraph. I am confused by use of the term apoptosis, which is defined as the process in which multicellular organisms destroy cells that are unnecessary or pose a threat. The term cannot be applied to a single-celled organism.

3) Results, second paragraph and legend to Figure 3. The fact that a heterotrophic bacterium cannot survive in a marine minimal medium without sources of C, N, and P is not up for debate. This paragraph seems simply to point this out, which is unnecessary.

4) Results, third paragraph. It has already been well established that members of this bacterial lineage are specialists on DMSP degradation and that Phaeobacter strains can grow on DMSP.

5) Results, third paragraph. Similar comments as in #3. Every bacterial strain must be provided with essential nutrients (C, N, P) in order to grow in culture. Since DMSP does not contain either nitrogen or phosphorus, an experiment showing that the organism won't grow solely on DMSP is reinventing the wheel.

6) Results, third paragraph. The methods say that 5.5 mM glucose was used as the carbon source in these experiments, so that's 33 mM carbon. DMSP was provided at 30 uM, which is 0.09 mM carbon. How could this experimental design be used to compare bacterial growth on glucose versus DMSP? I can't make sense of this.

7) Results, third paragraph. DMSP release by coccolithophores and its uptake by heterotrophic bacteria has long been established. I don't see how this fact constitutes evidence that DMSP is a chemical signal for colonization.

8) Results, fourth paragraph. This pattern could also be explained if the phytoplankton cultures crashed at exactly the same time, but the heterotrophic bacteria present in the co-culture degraded the dead/dying cells whereas the phytoplankton pure culture had no degradative activity.

9) Results, sixth paragraph. This paper provides a nice example of a marine bacterium that produces IAA and stimulates growth of phytoplankton. But make it clear here and elsewhere in the paper that this result has already been shown in other systems.

10) Results, tenth paragraph. Maybe. But it's also possible that the bacteria in the coculture simply take advantage of weakening phytoplankton cells that have already stopped growing, and that can't happen in the pure culture where degraders are not present. Distinguishing between these two possibilities is critical for determining whether this is indeed pathogenesis.

11) Results, eleventh paragraph. The gene expression data are misrepresented. The data in Table 1 and [Supplementary-material SD1-data] show that every gene tested except for one has higher expression in late stage cultures. The upregulated genes in the supplementary file include those indicating increasing phytoplankton growth rates, lipid biosynthesis, and DMS formation, as well as higher activity of carbon concentrating enzymes. This looks like selective reporting of data to fit the programmed cell death hypothesis.

12) Results, fourteenth paragraph. 0.1 mM tryptophan is a really high concentration, and must be 5 orders of magnitude higher than what can be measured in the ocean. I realize that concentrations will be higher in the zones where bacteria and phytoplankton cells are physically associated, but the requirement for concentrations that high suggest it might be a toxic effect or a metabolic imbalance in the coccolithophore.

14) Discussion, first paragraph. While this work shows growth promotion by the bacterium, it has not shown that death is initiated.

15) Discussion, first paragraph. I did not find convincing evidence of programmed cell death.

16) Discussion, fifth paragraph. I stumbled here and earlier in that paper over presentation of the fact that heterotrophic bacteria require sources of C, N, P and S for growth as a new finding. The important message from these experiments should be that the essential nutrients are all being delivered by a single phytoplankton species. This has also been shown several times before, however. Also, there is millimolar sulfur in seawater in the form of sulfate and almost all bacteria (including Phaeobacter) are able to reduce it. So, organic sulfur addition is not an essential component of the seawater medium.

*Reviewer #3:*

This is a very interesting and timely study examining the dynamic, small-scale interactions between a specific species of bacterium (Phaeobacter inhibens) and a globally important coccolithophore (*Emiliania huxleyi*), which oscillates between a stimulating mutualistic relationship to one what is more antagonistic in nature and results in cell death. The authors elegantly show using a combination of analytical approaches (that include growth physiology, microscopy, diagnostic staining/labeling, stable isotope labeling, mass spec analysis, manipulative experiments, and the use of auxotrophic mutants) that P. inhibits produces a phytohormone compound (indole-3-acetic acid, IAA) that serves as a growth enhancer for co-cultured *E. huxleyi* cells in this well-defined, cultured model system. IAA, in turn, derives from organic carbon exuded from *E. huxleyi* cells, and more specifically from tryptophan (the authors nicely use stable isotope labeling combined with mass spec to show this directly), the production of which is able to support growth for established tryptophan auxotrophs.

Co-habitating bacterial were also show to utilize DMSP (exogenously added and that derived from *E. huxleyi* cells) for enhanced growth and it's implied that the stimulatory impact by enhanced algal growth via IAA by residing bacteria is to stimulate production and access to DMSP as a source of organic sulfur. This has potentially important ecological and climatological impacts, given the role of DMS as a signaling bioactive molecule across orders of magnitude different size scales. Along these lines, the authors try to make this conceptual link by showing that Phaeobacter species are detected in natural *E. huxleyi* blooms (although the relative% of the Rhodobacteriaceae to other bacterial phyla among the ~2 million sequence reads of metagenomic data is not reported and would be useful to know in this context; see below) and that tryptophan is measured among the dissolved metabolites in natural *E. huxleyi* blooms.

IAA becomes physiologically toxic at high concentrations, which are achieved after ~17 d of co-culture and are density dependent, inducing a variety of cellular features of ROS-induced PCD including TUNEL positive cells, enhanced expression of various ROS and PCD-related genes, and elevated% of dead cells in co-cultures versus algal pure cultures. The activation of PCD is well known in *E. huxleyi* is well-documented in response to virus infection and other physiological stresses so this provides another mechanistic avenue for death transition for an important coccolithophore.

The scientific approach in this study was logical, direct, and very strong. The authors clearly established the phenomenon and systematically characterized the key metabolite players in the association. We are not only left with a quite detailed understanding of the dynamics of this ecophsysioloical interaction, but also are provided with detailed mechanistic landscape on what drives the interaction.

The mechanisms of action of how IAA stimulates tryptophan production in early-phase co-cultures and the PCD pathway in late-phase co-cultures are not known (or at least isn't reported here). These mechanistic and cell regulation aspects are interesting in terms of how this dynamic interaction has co-evolved with both stimulatory and destructive cell pathways.

Overall, this is a very strong paper and I recommend it for publication in *eLife* with minor revisions.

Some comments to address (some of which deal with some methodological details):

Abstract: the authors state that these interactions are likely to "impact micro-algal populations on a global scale". They very nicely show this interaction in culture and show some evidence that similar players are found in natural populations. However, extending it to a global scale is premature, given nothing is known of the relative and effective concentrations of these bacteria, IAA, tryptophan or associated dynamics with resident *E.huxleyi* natural populations. I agree that it poses some intriguing aspects as it relates to another potential bloom-termination mechanism (along with viruses & grazers) and the exciting part will be to now interrogate natural blooms for relative importance of this bacteria-mediated process in bloom termination.

Results, first paragraph: does the interaction with *E. huxleyi* preferentially occur with naked cells? This is an interesting observation, as all calcified cultures (or populations) do possess subpopulations of calcified and naked cells (which are easily differentiated with flow cytometry and side-scatter). Anything the authors can share here would provide important context as it relates to how the interaction may be structured in a population.

Results, last paragraph: I don't see this in the data presented in Figure 9. Can the authors elaborate? I only see a micrograph showing enhanced aggregation of the *E. huxleyi* cells.

Subsection “Monitoring bacterial growth in co-cultures”: I wonder if the authors can provide some rationale for why they used traditional plating techniques and CFU for assessing bacterial growth? I understand that this particular *P. inhibens* is culturable on 1/2YTSS plates. Are 1/2YTSS plates selective for this bacterium? Were other bacteria present in the co-culture experiments that necessitated its use? Did they every measure direct counts via DNA-intercolating stains coupled with microscopy or flow cytometry. I wonder if plating techniques were an accurate estimate of *P. inhibens* abundance given well- documented viable, but non-culturable states in marine microbiology. The use of plating/culturing is not routinely used in marine microbiology, given more direct methods are available.

Subsection “Fluorescent stains”, last paragraph: Looks like the authors forgot to include the relevant details for qualifying TUNEL positive cells.

Subsection “Gene expression analysis”; Table 1: I'd like more details on the relative expression analysis. Looks like they were done on duplicate cultures. Were any statistics done on those expression readings? Some of the columns are color-coded but no descriptive details are given.

Subsection “Roseobacticides lysis assay”: use of a diagnostic stain (e.g. SYTOX) to quantify lysis would be been useful here, instead of only relying on visual inspection.

Subsection “Metagenomic analysis”, last paragraph: It would help if the authors could provide a broader perspective on the metagenomic analysis. Were these total genome reads? Targeted 16S reads using tagged primers? What was the overall proportion of major bacterial phyla associated with the natural blooms? They only show things through the Rhodobacteraceae lens. How representative is it from a comparative% ? This helps provide some important ecological context.

[Editors' note: further revisions were requested prior to acceptance, as described below.]

Thank you for resubmitting your work entitled "Dynamic metabolic exchange governs a marine algal-bacterial interaction" for further consideration at *eLife*. Your revised article has been favorably evaluated by Detlef Weigel (Senior editor), a Reviewing editor and an outside reviewer.

The manuscript has been improved but there are some remaining issues that need to be addressed before acceptance, as outlined below:

*Reviewer #1:*

I have read the revised manuscript, and while it is better, it is still too long and redundant in several places. Re. the mechanism of enhancing growth by IAA – the authors should not use the analogy to higher plants for a single celled micro alga. Why not? Because in a micro alga, an increase in growth rate without some increase in light harvesting (or a decrease in cell carbon) implies an increase in quantum efficiency for growth. That doesn't make sense to me. I don't see what IAA would do to increase the quantum efficiency for growth. I don't think the authors really understand what IAA does that leads to the apparent increase in growth rate – and they probably should just state that rather than speculate using a model of a multicellular organism (a higher plant) that has roots and shoots, etc.

The basic idea of the experiments and their Discussion otherwise seems within bounds of reality – but, as stated above, there is far too much detail and redundancy for an average reader.

*Reviewer #2:*

Most of my concerns have been addressed.

One correction: Results and Discussion, second paragraph: Phaeobacter is unable to use nitrate as a nitrogen source. The bacterium therefore had no available nitrogen until ammonium was added, so this result is not surprising. The reference cited here (Zech et al., 2009) used ammonia in the medium.

---

## [Author Response]

[…]

*Summary:*

*After careful examination of your response to our request, we will reconsider this paper for possible publication in eLife. Please understand that the paper will be re-evaluated by the three original reviewers. To that end, I strongly suggest that the "global" importance of the findings be toned down – as the evidence from field studies about the "global" importance is lacking. That aspect of the paper was particularly irritating to one of the reviewers.*

*Reviewer #1:*

*This is a very thorough paper with many results indicating that a specific bacterium can stimulate growth of a coccolithophore and subsequently induce an apoptotic cascade. The key finding is that the signaling molecule is IAA = which is curious. IAA receptors in the algal cell clearly can enhance growth – but I don't understand why that happens. If the cells actually grew faster at the same irradiance, then the enhancement should alter the quantum yield for growth – which the authors do not report. One key parameter that appears to be missing to make that calculation is the optical absorption cross section normalized to chlorophyll (so-called a*). If the algal cell reduced the production of chlorophyll at a particular irradiance, the a* value would be higher (less self-absorption) and the cell could actually use light more efficiently. After longer exposure to IAA (or the association), it appears that plastids are becoming disrupted. If the authors have chlorophyll/cell values, the a* can potentially be calculated – and that would help give some rationale to the stimulation of growth.*

We thank reviewer #1 for this insightful comment; it certainly drove us to think deeper about the IAA effect on algal growth. To address the request to explore the mechanism underlying IAA-induced growth, we carried out chlorophyll a measurements in pure and co-cultures over a period of 17 days. These results are now included in our manuscript in Table 3. Chlorophyll a measurements did not provide strong insight regarding IAA-induced growth, and thus opened many more directions that could be explored. Therefore, we now discuss in detail the different possible mechanisms that could underlie algal growth enhancement by IAA. This discussion consists of three new paragraphs (Results and Discussion, paragraphs twenty to twenty-two).

*Reviewer #2:*

*There is some solid research here but it is spun into a story that is not fully supported by the data and, more importantly, much of the story was told a year ago in a paper in Nature. The new findings in the manuscript are the investigations of phenotypes of the phytoplankton and bacteria in response to IAA and tryptophan, but the concentrations needed to see the phenotypes are extremely high. This manuscript can be rewritten and shortened to highlight results that add to existing knowledge. But unfortunately, the study still will not address underlying mechanisms for phenotypic changes and will not constitute an advance sufficient for publication in eLife.*

We thank reviewer #2 for taking the time to thoroughly review our work. However, we disagree with this initial statement. Indeed IAA and tryptophan were reported a year ago by Amin et al. as signaling molecules acting between a diatom and a bacterium. Our study however, reveals how these signaling molecules act locally in the phycosphere underlying both mutualistic and antagonistic interactions, eventually leading to an apoptotic cascade. Our work reveals bacteria as key drivers of coccolithophore physiology, with DMSP, IAA, tryptophan and PCD demarcating the mechanistic landscape of the algal-bacterial interaction. The novelty of our work does not stem from reporting IAA and tryptophan as signaling molecules. Rather, it significantly builds beyond that important prior observation.

*1) The discovery that bacterially-produced indole-3-acetic acid serves as a growth stimulant to marine phytoplankton and that tryptophan release by the phytoplankton fuels IAA release has already been discovered and published (Amin et al., Nature, 2015). This manuscript provides evidence that another bacterium in the same taxonomic group (Roseobacter) has a similar relationship with a phytoplankton in a different group (coccolithophore), and thus adds evidence to an earlier finding. However, it does not represent a new discovery.*

The beautiful work by Amin et al. a year ago in Nature was indeed the first to report IAA and tryptophan crosstalk between a diatom and a bacterium. We found this work inspiring and cited it several times in our manuscript. However, in our work, IAA and tryptophan are reported as the metabolites linking coccolithophore population dynamics, bacterial colonization and the rapid algal demise that is driven by bacterially induced PCD. IAA and tryptophan are reported as the metabolites underlying these significant observations, thus expanding from the prior report.

In order to insure that the earlier report by Amin et al. 2015 is given appropriate credit in the context of our findings, in addition to existing citations of this work in our manuscript (Introduction, third paragraph; Results and Discussion, thirteenth and seventeenth paragraphs), we now further emphasize the previous discoveries through the following modifications in the Results and Discussion section:

“Intriguingly, the auxin indole-3-acetic acid (IAA) was previously demonstrated to play key roles in numerous terrestrial plant-bacteria interactions and was recently shown to be key in a marine association between a diatom and a Roseobacter bacterium”

“In line with this idea, in another algal-bacterial interaction it has been shown that tryptophan released by a diatom fuels IAA production by a bacterium (Amin et al., 2015).”

“Previously, tryptophan was identified as a key metabolite in the interaction between a diatom and a Roseobacter (Amin et al., 2015).”

We hope that the multiple citations of the work by Amin and colleagues and the mentioning of their findings in several places in our manuscript addressee the concerns raised by reviewer #2.

*2) Introduction, last paragraph. I am confused by use of the term apoptosis, which is defined as the process in which multicellular organisms destroy cells that are unnecessary or pose a threat. The term cannot be applied to a single-celled organism.*

In our use of the terms PCD and apoptosis we are really only standing on the shoulders of giants. The groundbreaking work that has revealed apoptosis and PCD in phytoplankton, including *E. huxleyi*, has provided solid evidence that these processes indeed occur in single cells. A very comprehensive review on the subject was recently published by Kay D. Bidle (Programmed Cell Death in Unicellular Phytoplankton, Current Biology, 2016) and is now referred to in our manuscript.

To assure that all readers are familiar with the concept of PCD in phytoplankton we have modified the text and added several citations for reference:

“Recent evidence suggests that environmental stresses and viral infection can trigger oxidative stress and a process similar to programmed cell death (PCD) in *E. huxleyi* (Bidle et al., 2007, Vardi et al., 2009, Bidle, 2016). The induction of PCD, which is an autocatalytic process, has been shown to occur in various widespread species of phytoplankton including *E. huxleyi* and functional links have been demonstrated between viral infection, PCD, and algal bloom collapse (Bidle, 2015, Bidle, 2016, Bidle and Vardi, 2011, Fulton et al., 2014, Vardi et al., 2012, Vardi et al., 2009, Rohwer and Thurber, 2009).”

*3) Results, second paragraph and legend to Figure 3. The fact that a heterotrophic bacterium cannot survive in a marine minimal medium without sources of C, N, and P is not up for debate. This paragraph seems simply to point this out, which is unnecessary.*

Since we aim to establish a robust model system in which the chemical exchange between microbes is elucidated, we strongly feel that this control experiment and the associated text are necessary. Because bacterial growth is later assessed upon addition of C, N, P and S, a negative control must be presented. Further, we feel that it is important to experimentally demonstrate that heterotrophic bacteria cannot utilize C, N, P and S as available in L1-Si and require them to be transformed by a primary producer to utilizable forms and/or concentrations. This paragraph has now been significantly modified to provide a clearer explanation to the results presented in Figure 3 (Results and Discussion, second paragraph).

*4) Results, third paragraph. It has already been well established that members of this bacterial lineage are specialists on DMSP degradation and that Phaeobacter strains can grow on DMSP.*

We thank reviewer #2 for pointing this out and have now added further citations to reinforce this matter: “It has been reported that Roseobacters and other bacteria can chemotax towards DMSP and catabolize it and various metabolic pathways for the bacterial use of the DMSP sulfur have been proposed and tracked (Miller and Belas, 2004, Miller et al., 2004, Seymour et al., 2010, Brock et al., 2013, Wang et al., 2016).”

Nevertheless, in order to establish a robust model system that allows examination of microbial DMSP fluxes amongst other infochemicals, we believe that bacterial DMSP consumption had to be demonstrated in our system under our experimental conditions.

*5) Results, third paragraph. Similar comments as in #3. Every bacterial strain must be provided with essential nutrients (C, N, P) in order to grow in culture. Since DMSP does not contain either nitrogen or phosphorus, an experiment showing that the organism won't grow solely on DMSP is reinventing the wheel.*

We failed to emphasize that the medium we are using (L1-Si) is made of seawater supplemented with N, P and S. Thus, when only DMSP is added in the experiment presented in Figure 3, added sources of N, P and S are already present in the medium. Specifically- 0.9 mM NaNO_3_, 0.04 mM NaH_2_PO_4_*H_2_O and 0.08 μM of NiSO_4_*6H_2_O, ZnSO_4_*7H_2_0 and CuSO_4_*5H_2_O. We have now included these details in the manuscript (Results and Discussion, first paragraph).

Our experiment showing that bacteria cannot grow solely on DMSP serves as a control demonstrating that while exuded DMSP can serve as a sulfur and to some extent a carbon source, bacteria still require phytoplankton to convert nitrogen and phosphorus to utilizable forms (even though they have been added to the medium originally). We now explain this matter in the text and discuss the necessity of the algal nutrient supply (Results and Discussion, second paragraph).

*6) Results, third paragraph. The methods say that 5.5 mM glucose was used as the carbon source in these experiments, so that's 33 mM carbon. DMSP was provided at 30 uM, which is 0.09 mM carbon. How could this experimental design be used to compare bacterial growth on glucose versus DMSP? I can't make sense of this.*

We thank the reviewer for pointing out the difference in carbon concentration between the treatments. Originally, since we detected only 6 μM DMSP accumulating in algal cultures we didn’t conduct the experiment in DMSP concentrations higher than 30 μM. To address the reviewers concern, we now tested bacterial growth in DMSP concentrations that provide comparable amounts of carbon. Indeed as commented by the reviewer, 5.5 mM glucose would supply 33 mM carbon. Since DMSP has 5 carbons, a DMSP concentration of 6 mM would supply 30 mM carbon. We now added experiments testing bacterial growth in 6 mM DMSP along with necessary controls and compare these results to the growth obtained in 5.5 mM glucose in Figure 3 and its legend. We mention this in the text as well: “Even when DMSP was added in higher concentrations to supply carbon in a comparable amount to the carbon supplied by the 5.5 mM glucose in the parallel experiments, bacterial growth was not evident (Figure 3 “NP + 6mM DMSP”).”

*7) Results, third paragraph. DMSP release by coccolithophores and its uptake by heterotrophic bacteria has long been established. I don't see how this fact constitutes evidence that DMSP is a chemical signal for colonization.*

Since *P. inhibens* can chemotact towards DMSP, possesses an adhesive organelle, and eventually is seen attached to algae – we suggest that exuded DMSP could serve as a colonization cue. Indeed, to prove this hypothesis further experiments are needed. We now highlighted the speculative nature of our statement in the text: “Thus, it is possible that DMSP serves as a chemical cue attracting bacteria to colonize the *E. huxleyi* host cell. Additional experiments are clearly needed to determine if indeed DMSP serves as an infochemical promoting bacterial colonization. Yet, it has been previously shown that algal DMSP and exudates serve as a strong cue to attract bacteria (Seymour et al., 2010, Smriga et al., 2016)”

*8) Results, fourth paragraph. This pattern could also be explained if the phytoplankton cultures crashed at exactly the same time, but the heterotrophic bacteria present in the co-culture degraded the dead/dying cells whereas the phytoplankton pure culture had no degradative activity.*

This is a very interesting point. However, judging by the results of our Sytox staining and TUNEL assay, when the co-culture is bleaching 90-94% of the algal population is dead. In the parallel pure culture at the same time only 21-35% are dead. So, death rates are very different between the cultures at the same time, indicating that bacteria promote algal death and not simply degrade cells that are already dying. Following the reviewer’s comment we now include this explanation: “One possible explanation for the bleaching observed is that the bacteria simply degrade dying algal cells at day 17 while similarly dying algal cells in the pure culture at the same time point remain intact. However, comparison of the death rates in the algal population in pure and co-cultures reveals a significant difference. At this time point, the vast majority of algal cells in the co-culture are dead (94% as indicated by Sytox staining, see Figure 3) while in the pure culture only 21% are dead by day 17. Thus, the presence of bacteria seems to play a key role in promoting algal death.”

*9) Results, sixth paragraph. This paper provides a nice example of a marine bacterium that produces IAA and stimulates growth of phytoplankton. But make it clear here and elsewhere in the paper that this result has already been shown in other systems.*

This seems to be a recurring concern and we apologize for not making it clear. We do not attempt to report IAA and tryptophan as novel signaling molecules, nor do we think this is the innovation and significance of our work. This section was modified: “Intriguingly, the auxin indole-3-acetic acid (IAA) was previously demonstrated to play key roles in numerous terrestrial plant-bacteria interactions and was recently shown to be key in a marine association between a diatom and a Roseobacter bacterium (Amin et al., 2015, Spaepen et al., 2007)”. To assure that the previous work is fully credited, below are listed all places in the manuscript where the work by Amin et al., 2015 is recognized:

1) Introduction, third paragraph

2) Results and Discussion, sixth paragraph

3) Results and Discussion, ninth paragraph

4) Results and Discussion, thirteenth paragraph

5) Results and Discussion, seventeenth paragraph

6) Results and Discussion, seventeenth paragraph

*10) Results, tenth paragraph. Maybe. But it's also possible that the bacteria in the coculture simply take advantage of weakening phytoplankton cells that have already stopped growing, and that can't happen in the pure culture where degraders are not present. Distinguishing between these two possibilities is critical for determining whether this is indeed pathogenesis.*

We believe that our data supports pathogenicity; higher mortality rates exemplified by Sytox green stain, transcription profiles indicating oxidative stress and PCD, and DNA fragmentation as the hallmark of PCD- all indicate that algae experience a bacterial-induced death. In order to clarify the collection of evidence supporting pathogenicity and in order to address the interesting point raised by reviewer #2 about weakening naked phytoplankton cells, we have now included an additional part in our Results and Discussion (twenty-third paragraph).

11) Results, eleventh paragraph. The gene expression data are misrepresented. The data in Table 1 and [Supplementary-material SD1-data] show that every gene tested except for one has higher expression in late stage cultures. The upregulated genes in the supplementary file include those indicating increasing phytoplankton growth rates, lipid biosynthesis, and DMS formation, as well as higher activity of carbon concentrating enzymes. This looks like selective reporting of data to fit the programmed cell death hypothesis.

Examining the maximal up-regulation of PCD-related genes (at day 17), shows that these genes were up-regulated between 2.75 to 6.81 fold. Other examined genes (presented in the supplementary file), at the same time point, show expression ratios between 0.5 and 3.48, with only three genes actually up-regulated more than 2.75 fold (the lowest up-regulation value for PCD genes). We have now added statistical analysis to strengthen the presentation of the difference in expression levels between the two gene groups: “To further explore the death process experienced by algae in co-culture, we examined the expression profile of a select group of algal genes. […] The resulting probability of the T-test was 2.8 x 10^-11^ indicating that the difference between the two datasets is statistically significant.”

*12) Results, fourteenth paragraph. 0.1 mM tryptophan is a really high concentration, and must be 5 orders of magnitude higher than what can be measured in the ocean. I realize that concentrations will be higher in the zones where bacteria and phytoplankton cells are physically associated, but the requirement for concentrations that high suggest it might be a toxic effect or a metabolic imbalance in the coccolithophore.*

This is a very important point raised by reviewer #2. This is the reason we conducted a control experiment in which pure algal cultures were supplemented with the same concentration of tryptophan. The pure cultures did not present accelerated death. These results, along with the hyper-colonization phenotype of bacteria indicate that accelerated algal death in the presence of added tryptophan, was mediated by bacteria.

In light of this comment, we have modified the legend of Figure 9: “(a) Images of co-cultures and algal cultures at day 10. Upon inoculation of bacteria, 0.1 mM tryptophan (trp) was added. Seen is a co-culture that bleached a week earlier (middle). An algal culture treated with the same concentration of tryptophan did not display bleaching (right).”

*13) Discussion, first paragraph. While this work shows growth promotion by the bacterium, it has not shown that death is initiated.*

*14) Discussion, first paragraph. I did not find convincing evidence of programmed cell death.*

In response to both of these comments we now include and discuss the earlier pioneering work that established PCD in phytoplankton, introducing the TUNEL assay as a tool to assess apoptosis in single cells. In our work, the combination of higher death rates, gene expression data, and DNA fragmentation assessed by the TUNEL assay clearly demonstrate that bacteria induce algal PCD. These results adhere to the rigorous standard previously established by leading groups that revealed PCD in phytoplankton. The text was modified to assure al readers are introduced with the concept of PCD in phytoplankton and the established use of the TUNEL assay: “PCD naturally occurs in an aging algal population and can be triggered by a variety of biotic stresses such as viral infection and abiotic environmental stresses such as nutrient limitation and various light regimes (Bidle, 2015, Bidle, 2016). […] Using the TUNEL assay (see Materials and methods) we could detect that 90% of algal cells in 20-day old co-cultures contain highly fragmented DNA in comparison to 35% in axenic algal cultures of the same age (Figure 6).”

*15) Discussion, fifth paragraph. I stumbled here and earlier in that paper over presentation of the fact that heterotrophic bacteria require sources of C, N, P and S for growth as a new finding. The important message from these experiments should be that the essential nutrients are all being delivered by a single phytoplankton species. This has also been shown several times before, however. Also, there is millimolar sulfur in seawater in the form of sulfate and almost all bacteria (including Phaeobacter) are able to reduce it. So, organic sulfur addition is not an essential component of the seawater medium.*

We thank the reviewer for their suggestion on how to present these observations with more clarity. We have significantly modified the sections concerning nutrient utilization and hope that these modifications will address the reviewer’s concerns. Sections discussing nutrient utilization include:

Results and Discussion, first and second paragraphs.

*Reviewer #3:*

[…]

*Some comments to address (some of which deal with some methodological details):*

*Abstract: the authors state that these interactions are likely to "impact micro-algal populations on a global scale". They very nicely show this interaction in culture and show some evidence that similar players are found in natural populations. However, extending it to a global scale is premature, given nothing is known of the relative and effective concentrations of these bacteria, IAA, tryptophan or associated dynamics with resident E. huxleyi natural populations. I agree that it poses some intriguing aspects as it relates to another potential bloom-termination mechanism (along with viruses & grazers) and the exciting part will be to now interrogate natural blooms for relative importance of this bacteria-mediated process in bloom termination.*

Indeed, extending our findings to global scale dynamics is pre-mature. We have revised our statements and rephrased them to make it clear that future interrogation is necessary to determine the importance and magnitude of our findings in the ocean: “Coccolithophore-bacteria interactions should be further studied in the environment to reveal whether they impact micro-algal population dynamics on a global scale.”

*Results, first paragraph: does the interaction with E. huxleyi preferentially occur with naked cells? This is an interesting observation, as all calcified cultures (or populations) do possess subpopulations of calcified and naked cells (which are easily differentiated with flow cytometry and side-scatter). Anything the authors can share here would provide important context as it relates to how the interaction may be structured in a population.*

This is an intriguing subject that indeed merits further discussion in the manuscript. The interaction we observed occurs solely with naked algal cells. Whether bacteria target cells that are already naked (suggesting sub-populations) or trigger coccoliths shedding is unknown. We now added a new paragraph in our Discussion to address this matter: “Bacteria in our co-cultures were found attached solely to naked algal cells. We did not detect bacteria attached to calcified algae or shed coccoliths. What underlies the specific bacterial attachment to naked algal cells? […] In this regard, the physical properties of a naked non-calcified algal cell might be more suitable for a bacterial attachment. Similar influences might promote the permanent attachment that persists even after the algal cell has died (Figure 10).”

*Results, last paragraph: I don't see this in the data presented in Figure 9. Can the authors elaborate? I only see a micrograph showing enhanced aggregation of the E. huxleyi cells.*

To address this comment we have now included an inset of higher magnification (Figure 9) and added the following text to the legend of Figure 9: “Note that in the tryptophan treated co-culture (+trp) in (c) each algal cell has attached bacteria and the bacterial coverage is so dense that several covered algal cells cannot be seen”.

*Subsection “Monitoring bacterial growth in co-cultures”: I wonder if the authors can provide some rationale for why they used traditional plating techniques and CFU for assessing bacterial growth? I understand that this particular P. inhibens is culturable on 1/2YTSS plates. Are 1/2YTSS plates selective for this bacterium? Were other bacteria present in the co-culture experiments that necessitated its use? Did they every measure direct counts via DNA-intercolating stains coupled with microscopy or flow cytometry. I wonder if plating techniques were an accurate estimate of P. inhibens abundance given well- documented viable, but non-culturable states in marine microbiology. The use of plating/culturing is not routinely used in marine microbiology, given more direct methods are available.*

1/2YTSS plates are not selective for *P. inhibens*, but certainly support its growth. Since pure cultures of *P. inhibens* were used to inoculate axenic algal cultures, only *P. inhibens* bacteria were expected to grow. Thus, monitoring via CFU was selected as the method of choice. As indicated in the subsection “Monitoring bacterial growth in co-cultures”, while this method is convenient, it might underrepresent bacterial numbers due to the presence of multi-cellular rosettes.

*Subsection “Fluorescent stains”, last paragraph: Looks like the authors forgot to include the relevant details for qualifying TUNEL positive cells.*

This section was revised and now includes the following details: “Positive controls were generated by pretreating cells with 15 U ml ^-1^ DNase I (Thermo Fischer Scientific). These controls yielded a very high fluorescent signal and thus were scaled differently to avoid image saturation.”

*Subsection “Gene expression analysis”; Table 1'd like more details on the relative expression analysis. Looks like they were done on duplicate cultures. Were any statistics done on those expression readings? Some of the columns are color-coded but no descriptive details are given.*

To address the reviewer’s comments, we now added a detailed explanation in the Materials and methods section regarding the NanoString expression data: “Experiments were conducted in duplicates and gene expression data from all experiments was normalized as one dataset. […] Background expression was determined as mean+(2*standard deviation), and expression data lower than this value were discarded.”

We also added a table (see [Supplementary-material SD3-data]) containing the average expression values in each experiment (Average), the standard deviation (StDev), a representation of the standard deviation as percent from the average value (StDev% ), and all ratios presented in the manuscript.

*Subsection “Roseobacticides lysis assay”: use of a diagnostic stain (e.g. SYTOX) to quantify lysis would be been useful here, instead of only relying on visual inspection.*

In addition to the inspection under the microscope, no bleaching was detected. While the use of Sytox would reveal accurate mortality rates, no obvious phenotype seemed to justify further exploration. The lack of bleaching is now mentioned in the manuscript (Results and Discussion, twelfth paragraph).

*Subsection “Metagenomic analysis”, last paragraph: It would help if the authors could provide a broader perspective on the metagenomic analysis. Were these total genome reads? Targeted 16S reads using tagged primers? What was the overall proportion of major bacterial phyla associated with the natural blooms? They only show things through the Rhodobacteraceae lens. How representative is it from a comparative% ? This helps provide some important ecological context.*

We apologize for the partial presentation of our Metagenomic analysis. Both reviewer #3 and #2 have commented on this matter and we now address their concerns in our analysis description.

Specifically, relevant sections in Materials and methods have been modified (subsection “Metagenomic analysis”), and the legend of Figure 1 was modified as well: “Metagenomic analysis of Roseobacters associated with *E. huxleyi* blooms reveals co-occurrence of *P. inhibens.* Two *E. huxleyi* blooms were sampled in the Gulf of Maine during the summer of 2015 and metagenomic analysis of the bacterial population was performed (see Materials and methods). Shown is the relative abundance of members of the Rhodobacteraceae family, which accounted for 6% of bacteria. The members of the Rhodobacteraceae family were detected in both blooms and their abundance changed ± 2% between replicates and between the two blooms. *P. inhibens* was present in both blooms and is indicated by an asterisk. Shown are the results for the July 2015 bloom (see Materials and methods).”

[Editors' note: further revisions were requested prior to acceptance, as described below.]

*Reviewer #1:*

*I have read the revised manuscript, and while it is better, it is still too long and redundant in several places. Re. the mechanism of enhancing growth by IAA – the authors should not use the analogy to higher plants for a single celled micro alga. Why not? Because in a micro alga, an increase in growth rate without some increase in light harvesting (or a decrease in cell carbon) implies an increase in quantum efficiency for growth. That doesn't make sense to me. I don't see what IAA would do to increase the quantum efficiency for growth. I don't think the authors really understand what IAA does that leads to the apparent increase in growth rate – and they probably should just state that rather than speculate using a model of a multicellular organism (a higher plant) that has roots and shoots, etc.*

*The basic idea of the experiments and their Discussion otherwise seems within bounds of reality – but, as stated above, there is far too much detail and redundancy for an average reader.*

Reviewer #1 is indeed correct; we do not understand how IAA stimulates increased algal yields. We now clearly state it in the text. We have removed the entire paragraph discussing IAA signaling in plants to address this comment (Results and Discussion, nineteenth paragraph).

To address the concern of reviewer #1 regarding length and redundancy we have removed and significantly shortened multiple paragraphs in our manuscript.

*Reviewer #2:*

*Most of my concerns have been addressed.*

*One correction: Results and Discussion, second paragraph: Phaeobacter is unable to use nitrate as a nitrogen source. The bacterium therefore had no available nitrogen until ammonium was added, so this result is not surprising. The reference cited here (Zech et al., 2009) used ammonia in the medium.*

We thank reviewer #2 for this comment. We have modified this paragraph to clearly state that the addition of utilizable forms of nutrients (i.e. ammonium) did not increase bacterial growth unless all major nutrients were added (Results and Discussion, second paragraph).